# Brain network science modelling of sparse neural networks enables Transformers and LLMs to perform as fully connected

**Yingtao Zhang**[1,2]**, Diego Cerretti**[1,2]**, Jialin Zhao**[1,2]**, Ziheng Liao**[1,2]**, Wenjing Wu**[1,2]
**Umberto Michieli**[4,5] **& Carlo Vittorio Cannistraci**[1,2,3*]

[1]Center for Complex Network Intelligence (CCNI)[†]
[2]Dept. of Computer Science & Technology, [3]School of Biomedical Engineering, Tsinghua University
[4]University of Padova, [5]Canva Research

## Abstract

This study aims to enlarge our current knowledge on the application of brain-inspired network science principles for training artificial neural networks (ANNs) with sparse connectivity. Dynamic sparse training (DST) emulates the synaptic turnover of real brain networks, reducing the computational demands of training and inference in ANNs. However, existing DST methods face difficulties in maintaining peak performance at high connectivity sparsity levels. The Cannistraci-Hebb training (CHT) is a brain-inspired method that is used in DST for growing synaptic connectivity in sparse neural networks. CHT leverages a gradient-free, topology-driven link regrowth mechanism, which has been shown to achieve ultra-sparse (1% connectivity or lower) advantage across various tasks compared to fully connected networks. Yet, CHT suffers two main drawbacks: (i) its time complexity is $\mathcal{O}(N \cdot d^3)$- N node network size, d node degree - hence it can be efficiently applied only to ultra-sparse networks. (ii) it rigidly selects top link prediction scores, which is inappropriate for the early training epochs, when the network topology presents many unreliable connections. Here, we design the first brain-inspired network model - termed bipartite receptive field (BRF) - to initialize the connectivity of sparse artificial neural networks. Then, we propose a matrix multiplication GPU-friendly approximation of the CH link predictor, which reduces the computational complexity to $\mathcal{O}(N^3)$, enabling a fast implementation of link prediction in large-scale models. Moreover, we introduce the Cannistraci-Hebb training soft rule (CHTs), which adopts a flexible strategy for sampling connections in both link removal and regrowth, balancing the exploration and exploitation of network topology. Additionally, we propose a sigmoid-based gradual density decay strategy, leading to an advanced framework referred to as CHTss. Empirical results show that BRF offers performance advantages over previous network science models. Using 1% of connections, CHTs outperforms fully connected networks in MLP architectures on visual classification tasks, compressing some networks to less than 30% of the nodes. Using 5% of the connections, CHTss outperforms fully connected networks in two Transformer-based machine translation tasks. Finally, with only 30% of the connections, both CHTs and CHTss achieve superior performance over other dynamic sparse training methods, and perform on par with—or even surpass—their fully connected counterparts in language modeling across various sparsity levels within the LLaMA model family.

---

[*]Corresponding author: `kalokagathos.agon@gmail.com`

[†]Research Center in Tsinghua Laboratory of Brain and Intelligence (THBI), Department of Psychological and Cognitive Sciences.

39th Conference on Neural Information Processing Systems (NeurIPS 2025).

The code is available at: `https://github.com/biomedical-cybernetics/Cannistraci-Hebb-Training-Soft-Rule-`.

# 1 Introduction

Artificial neural networks (ANNs) have led to significant advances in various fields such as natural language processing, computer vision, and deep reinforcement learning. The most common ANNs consist of several fully connected (FC) layers, which account for a large portion of the total parameters in recent large language models [1, 2]. This dense connectivity poses major challenges during the model training and deployment phases. In contrast, neural networks in the brain inherently exhibit sparse connectivity [3, 4]. This natural design in the brain exploits sparsity, suggesting a model in which the number of connections does not scale quadratically with the number of neurons. This could alleviate computational constraints, enabling more scalable network architectures.

Dynamic sparse training (DST) [5, 6, 7, 8, 9] has emerged as a promising approach to reduce computational and memory overhead of training deep neural networks while maintaining or even improving model performance. DST is also biologically inspired: it draws an analogy to synaptic turnover [10] in the brain, a fundamental neurobiological process in which synapses are continuously formed, strengthened, weakened, and eliminated over time. This dynamic rewiring enables the brain to adapt, learn, and store memories efficiently while preserving the overall stability of the network. Similarly, in DST, connections are dynamically pruned and regrown throughout training, allowing the network to adapt its connectivity structure in response to learning signals while maintaining a fixed sparsity level.

Apart from some detailed distinctions, the primary innovation in this field centers on the development of the regrowth criterion. A notable advancement is the gradient-free regrowth method introduced by Cannistraci-Hebb training (CHT) [9]. This method is inspired by epitopological learning—literally meaning 'new topology'—and is rooted in brain-inspired network science theory [11, 12, 13, 14, 15]. Epitopological learning explores how learning can be implemented on complex networks by changing the shape of their connectivity structure (epitopological plasticity). CHT has demonstrated remarkable advantages in training ultra-sparse ANNs with connectivity levels of 1% or lower, often outperforming fully connected networks in various tasks. However, despite these advances, CHT encounters two major challenges: 1) During dynamic sparse training, its rigid link selection mechanism can lead to *epitopological local minima* where the sets of removed links and regrown links exhibit significant overlap, severely impedes the exploration of optimal network topologies. 2) The time complexity of the CHT regrowth method, Cannistraci-Hebb 3 on Length 3 paths (CH3-L3p), is $\mathcal{O}(N \cdot d^3)$, where $N$ represents the number of nodes in the network and $d$ is the average degree. A length 3 path is a walk of three consecutive links. As the network becomes denser, the time complexity approaches $\mathcal{O}(N^4)$, rendering it impractical for large-scale and higher-density models.

In this article, we present the **C**annistraci-**H**ebb **T**raining **s**oft rule (CHTs), which introduces several key innovations: 1) To address the issue of epitopological local minima, CHTs employs a multinomial distribution to sample link scores from removal and regrowth metrics, enabling more flexible and effective exploration of network topologies. 2) CHTs incorporates novel substitution node-based link prediction mechanisms, reducing the computational time complexity to $\mathcal{O}(N^3)$. This improvement makes CHTs scalable to large-scale models with higher network density. 3) CHTs initializes the sparse topologies for bipartite networks with the brain-like receptive field, demonstrating superior performance compared to the traditional Erdős–Rényi [5], bipartite small world, and bipartite scale-free model [9]. Additionally, we propose a sigmoid gradual density decay strategy, which, when integrated with CHTs, forms an enhanced framework termed CHTss. This combined approach further optimizes the training process for sparse neural networks.

To evaluate the effectiveness of the **C**annistraci-**H**ebb **T**raining **s**oft rule with **s**igmoid density decay (CHTss), we conduct extensive experiments across multiple architectures and tasks. Firstly, to assess the basic concept of CHTs, we employ MLPs on benchmark datasets, including MNIST [16], EMNIST [17], and Fashion MNIST [18]. The results show that CHTs performs better than fully connected networks with only 1% of the connections (99% sparsity) in MLPs. Further, to evaluate the end-to-end approach CHTss, we utilize Transformers [19] on machine translation datasets such as Multi30k en-de [20], IWSLT14 en-de [21], and WMT17 en-de [22]. From the experimental results, CHTss surpasses fully connected Transformers while using only 5% of the connections

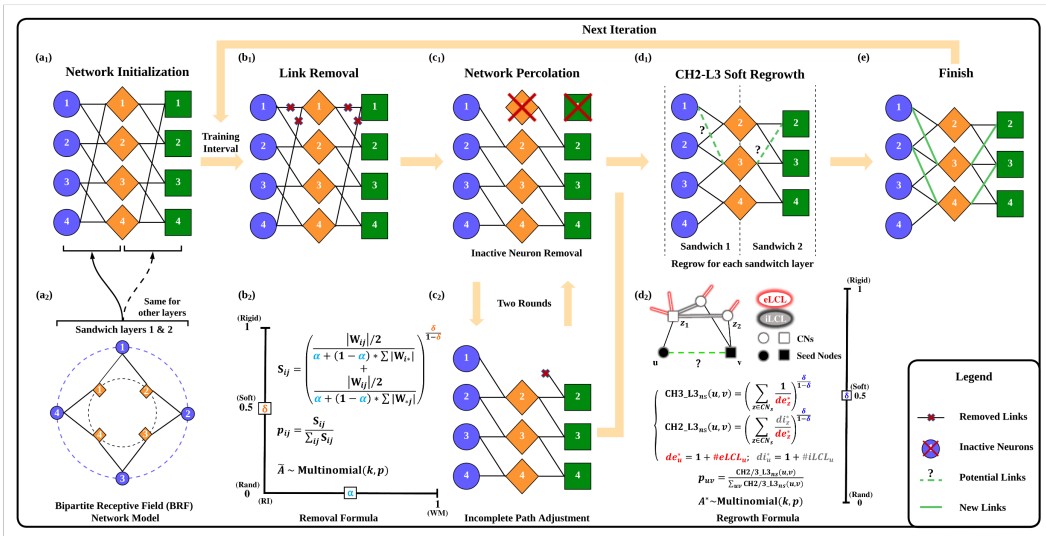

Figure 1: **Illustration of the CHTs process.** One training iteration follows the steps of (a1) → (b1) → (c1) → (c2) → (d1) → (e). (a1) Network initialization with each of the sandwich layers (bipartite networks connecting layers' input nodes to their output nodes) being a bipartite receptive field (BRF) network. (a2) BRF network representation with $r = 0$. (b1) Link removal process. (b2) Formula for determining which links to remove. (c1) Removal of inactive neurons caused by link removal. (c2) Adjust and remove incomplete links caused by inactive neuron removal. (d1) Regrowth of links according to the CH2-L3 node-based soft rule. (d2) Detailed illustration of the CH2-L3 node-based soft rule. (e) Finished state of the network after one iteration. The next iteration repeats the steps (b1) - (e) from this finished state. $\tilde{A}$ indicates the removal set of the iteration and $A^*$ is the regrown set.

on Multi30k and IWSLT, and achieves performance comparable to fully connected LLaMA-60M and LLaMA-130M models on OpenWebText language modeling tasks. Moreover, with 30% of the connections, CHTs even outperforms the fully connected LLaMA-1B counterpart. These findings underscore the potential of CHTs and CHTss in enabling highly efficient and effective large-scale sparse neural network training.

## 2   Related Work

### 2.1   Dynamic sparse training

Dynamic sparse training is a subset of sparse training methodologies. Unlike static sparse training methods (also known as pruning at initialization) [23, 24, 25, 26], dynamic sparse training allows for the evolution of network topology during the training process. The pioneering method in this field is Sparse Evolutionary Training (SET) [5], which removes links based on the magnitude of their weights and regrows new links randomly. Subsequent developments have sought to refine and expand upon this concept of dynamic topological evolution. One such advancement was proposed by DeepR [27], a method that adjusts network connections based on stochastic gradient updates combined with a Bayesian-inspired update rule. Another significant contribution is RigL [7], which leverages the gradient information of non-existing links to guide the regrowth of new connections during training. MEST [8] utilizes both gradient and weight magnitude information to selectively remove and randomly regrow new links, analogously to SET. In addition, it introduces an EM&S strategy that allows the model to train at a higher density and gradually converge to the target sparsity. The Top-KAST [6] method maintains constant sparsity throughout training by selecting the top $K$ parameters based on parameter magnitude at each training step and applying gradients to a broader subset $B$, where $B \supset A$. To avoid settling on a suboptimal sparse subset, Top-KAST also introduces an auxiliary exploration loss that encourages ongoing adaptation of the mask. Additionally, sRigL [28] adapts the principles of RigL to semi-structured sparsity, facilitating the training of vision models from scratch with actual speed-ups during training phases. Despite these advancements, the state-of-the-art method remains RigL-based, yet it is not fully sparse in backpropagation, necessitating

the computation of gradients for non-existing links. Addressing this limitation, Zhang et al. [9] propose CHT, a dynamic sparse training methodology that adopts a gradient-free regrowth strategy that relies solely on topological information (network shape intelligence), achieving an ultra-sparse configuration that surpasses fully connected networks in some tasks. Some extra related works of DST are provided in Appendix M.

## 2.2 Cannistraci-Hebb Theory and Network Shape Intelligence

As the SOTA gradient-free link regrown method, CHT [9] originates from a brain-inspired network science theory. Drawn from neurobiology, Hebbian learning was introduced in 1949 [29] and can be summarized in the axiom: "neurons that fire together wire together." This could be interpreted in two ways: changing the synaptic weights (weight plasticity) and changing the shape of synaptic connectivity [11, 12, 13, 14, 15]. The latter is also called *epitopological plasticity* [11] because plasticity means "to change shape," and *epitopological* means "via a new topology." *Epitopological Learning* (EL) [12, 13, 14] is derived from this second interpretation of Hebbian learning and studies how to implement learning on networks by changing the shape of their connectivity structure. One way to implement EL is via link prediction, which predicts the existence and likelihood of each nonobserved link in a network. CH3-L3 is one of the best-performing and most robust network automata, belonging to the Cannistraci-Hebb (CH) theory [30], which can automatically evolve the network topology starting from a given structure. The rationale is that, in any complex network with local-community organization, the cohort of nodes tends to be co-activated (fire together) and to learn by forming new connections between them (wire together) because they are topologically isolated in the same local community [30]. This minimization of the external links induces a topological isolation of the local community, which is equivalent to forming a barrier around it. The external barrier is fundamental to maintaining and reinforcing the signaling in the local community, inducing the formation of new links that participate in epitopological learning and plasticity.

## 3 Cannistraci-Hebb training soft rule with sigmoid gradual density decay

### 3.1 Soft removal and regrowth.

> **Definition 1. Epitopological local minima.** In the context of dynamic sparse training methods, we define an epitopological local minima (ELM) as a state where the sets of removed links and regrown links exhibit a significant overlap.

Let $A_t$ be the set of existing links in the network at the training step $t$. Let $\tilde{A}_t$ be the set of removal links and $A_t^*$ be the set of regrown links. The overlap set between removed and regrown links at step $t$ can be quantified as $O_t = \tilde{A}_t \cap A_t^*$. An ELM occurs if the size of $O_t$ at step $t$ is significantly large compared to the size of $A_t^*$, indicating a high probability of the same links being removed and regrown repeatedly throughout the subsequent training steps. This can be formally represented as $\frac{|O_t|}{|A_t^*|} \geq \theta$, where $\theta$ is a predefined threshold close to 1, indicating strong overlap. This definition is essential for the understanding of CHT, as evidenced by the article [9] indicating that the overlap rate between removed and regrown links becomes significantly high within just a few epochs, leading to rapid topological convergence towards the ELM. Previously, CHT implemented a topological early stop strategy to avoid predicting the same links iteratively. However, it will stop the topological exploration very fast and potentially trap the model within the ELM.

In this article, we adopt a probabilistic approach where the regrowth process is modeled as sampling from a $\{0, 1\}$ multinomial distribution, with probabilities determined by link prediction scores, thereby introducing a "soft sampling" mechanism. Under this formulation, each mask value is not rigidly dictated by the link prediction score; instead, low-score links can still be selected with lower probability, facilitating escape from epitopological local minima (ELM).

To demonstrate that soft sampling effectively balances exploration and exploitation, we evaluate its behavior in Figure 3, which presents the impact of varying softness levels during training of LLaMA-60M for 5000 steps under 90% sparsity. We compare the in-time over-parameterization (ITOP) rate [31], which quantifies the cumulative proportion of links activated throughout training. As shown, deterministic regrowth leads to rapid topological convergence after approximately 1000

steps, indicating that it becomes trapped in an ELM without further exploration. Random regrowth, while capable of escaping ELMs by introducing new connections, fails to exploit the underlying topological structure effectively. In contrast, soft regrowth achieves a balance by both exploiting the current topology and exploring new link combinations probabilistically. This balance enables a more appropriate exploration of the connectivity space, ultimately leading to superior performance, as evidenced by the results.

**Link removal alternating from weight magnitude and relative importance.** We illustrate the link removal part of CHTs in Figure 1b1) and b2). We employ two methods, Weight Magnitude (WM) $|\mathbf{W}|$ and Relative Importance (RI) [32], to remove the connections during dynamic sparse training. Given an input node $i$ and an output node $j$ connected with weight $W_{ij}$, we define the relative importance $RI_{ij}$ as:

$$\mathbf{RI}_{ij} = \frac{|\mathbf{W}_{ij}|}{\sum |\mathbf{W}_{*j}|} + \frac{|\mathbf{W}_{ij}|}{\sum |\mathbf{W}_{i*}|} \tag{1}$$

As illustrated in Equation 1, RI assesses connections by normalizing the absolute weight of links that share the same input or output neurons. This method does not require calibration data and can perform comparably to the baseline post-training pruning methods like sparsegpt [33] and wanda [34]. Generally, WM and RI are straightforward, effective, and quick to implement in DST for link removal, but give different directions for network percolation. WM prioritizes links with higher weight magnitudes, leading to rapid network percolation, whereas RI inherently values links connected to lower-degree nodes, thus maintaining a higher active neuron post-percolation (ANP) rate. The ANP rate is the ratio of the number of active neurons after training over the original number of neurons before training. These methods are equally valid but cater to different scenarios. For instance, using RI significantly improves results on the Fashion MNIST dataset compared to WM, whereas WM performs better on the MNIST and EMNIST datasets.

**Soft link removal.** In the early stages of training, both WM and RI are not reliable due to the model's underdevelopment. Therefore, rather than strictly selecting top values based on WM and RI, we also sample links from a multinomial distribution using an importance score calculated by the removal metrics. The final formula for link removal is defined in Equation 2.

$$\mathbf{S}_{ij} = \left( \frac{|\mathbf{W}_{ij}|/2}{\alpha + (1-\alpha) \sum |\mathbf{W}_{i*}|} + \frac{|\mathbf{W}_{ij}|/2}{\alpha + (1-\alpha) \sum |\mathbf{W}_{*j}|} \right)^{\frac{\delta}{1-\delta}} \tag{2}$$

Here, $\alpha$ determines the removal strategy, shifting from weight magnitude ($\alpha = 1$) to relative importance ($\alpha = 0$). In all experiments, we only evaluate these two $\alpha$ values. $\delta$ adjusts the softness of the sampling process. As training progresses and weights become more reliable, we adaptively increase $\delta$ from 0.5 to 0.75 to refine the sampling strategy and improve model performance. These settings are constant for all the experiments in this article.

**Node-based link regrowth.** Another significant challenge for CHT lies in the time complexity of link prediction. In the original CHT framework [9], the path-based CH3-L3p metric is employed for link regrowth, as discussed in Appendix C. However, this method incurs a high computational cost due to the need to compute and store all length-three paths, resulting in a time complexity of $O(N \cdot d^3)$, where $N$ is the number of nodes and $d$ is the network's average degree. This complexity is prohibitive for large models with numerous nodes and higher-density layers.

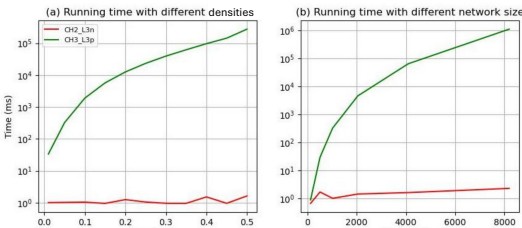

Figure 2: **One-time Link Prediction Runtime Performance Evaluation** of node-based and path-based methods across varying densities and network sizes. In (a), the network size is fixed at 1024 × 1024, while in (b), the density is fixed at 5%.

To address this issue, we introduce a more efficient, node-based paradigm that eliminates the reliance on length-three paths between seed nodes. Instead, this approach focuses on the

common neighbors of seed nodes. The node-based version of CH3-L3p, denoted as CH2-L3n, also depends on the internal local community links (iLCL, the links between the common neighbors [30]) to enhance the expressiveness of the formula. This variant is formulated as:

$$\textbf{CH2-L3n}(u,v) = \sum_{z \in L3} \frac{di_z^*}{de_z^*} \tag{3}$$

Here, $u$ and $v$ denote the seed nodes, while $z$ is the common neighbor on the L3 path [30], a walk of three consecutive links that connects $u$ to $v$ via two of those intermediate nodes. The terms $di_z^*$ $de_z^*$ represent the number of internal local community links (iLCL) and external local community links (eLCL) of node $i$, with a default increment of 1 to prevent division by zero. The detailed explanation of iLCL and eLCL can be found in Appendix C. The pseudocode is provided in the supplementary material. The new node-based variant, CH2-L3n, reduces the computational complexity to $O(N^3)$ and enables efficient matrix-based computations on GPUs. We evaluate the one-time link prediction runtime performance of both CH3-L3p and CH2-L3n across different network sizes and sparsity levels, as illustrated in Figure 2. The red and green lines depict the actual runtime for the path-based and node-based methods, respectively. The results reveal that the node-based version achieves significantly faster runtime performance compared to the path-based methods. Furthermore, the node-based method demonstrates consistently stable runtime across diverse network sizes and density levels, making it an ideal link prediction as the CH theory-based link regrowth component of CHTs in dynamic sparse training for large-scale models.

## 3.2 Bipartite receptive field network modeling.

In this article, we propose the Bipartite Receptive Field (BRF) network model, a novel sparse topological initialization method capable of generating brain-network-like receptive field connectivity. The principal topological initialization approaches for dynamic sparse training are grounded in network science theory, where three basic generative models for monopartite sparse artificial complex networks are the Erdős-Rényi (ER) model [35], the Watts-Strogatz (WS) model [36], and the Barabási-Albert (BA) model [37], which are not brain-inspired. Since the standard WS and BA models are not directly designed for bipartite networks, they were recently extended [9] into their bipartite counterparts and term as Bipartite Small-World (BSW) and Bipartite Scale-Free (BSF), respectively. BSW generally outperforms BSF for dynamic sparse training (see Appendix D).

During BSW initialization, each output node is connected to its spatially nearest input nodes. This spatially local connectivity pattern aligns with the concept of receptive fields observed in biological neural systems, where neurons respond selectively to localized regions of input space. However, the rewiring process of BSW does not follow brain mechanisms: it simply deletes a set of links from the closer input nodes to rewire them uniformly at random anywhere on the input layer. Conversely, the random allocation of connectivity in brain network topologies is guided by the spatial distance of the neurons [38, 39]. Unlike the BSW model that introduces random connectivity by a rewiring process, which cannot control the extent of spatial-dependent randomness injected in the topology, the BRF model directly generates a connectivity with a customized level of spatial-dependent randomness using a parameter $r \in [0, 1]$. A low value of $r$ results in links that are densely clustered around the diagonal, while a higher value of $r$ leads to less clustered connectivity patterns, which tend to be uniformly at random only for $r = 1$. Specifically, a bipartite adjacency matrix with links near the diagonal indicates that adjacent nodes from the two layers are linked, whereas links far from the diagonal correspond to more distant node pairs. The mathematical formula and implementation are detailed in Appendix D.

Furthermore, the degree distribution of the BSW model is fixed to the same value for all the nodes at the same layer before rewiring, whereas after rewiring, the degree distribution is not conserved, and the more links it rewires, the more it will be similar to the ER model. Instead, the BRF model has the important property to conserve the degree distribution of the output layer, which ensures that it maintains a designed receptive field connectivity. This means that an initialization setting of the BRF model is the output degree distribution, which in this study we consider fixed or uniformly at random, as shown in Appendix D. We also conduct a sensitivity test of the influence of $r$ in Figure 7a). It should be noted that when $r = 0$, the network is equivalent to the BSW with $\beta = 0$, and when $r = 1$, the network becomes an ER network. The examples of the adjacency matrices of BSF, BSW, and BRF are shown in Figure 5.

### 3.3 Sigmoid Gradual Density Decay

As demonstrated in GraNet [40] and $\text{MEST}_{EM\&S}$ [8], incorporating a density decrease strategy can significantly improve the performance of dynamic sparse training. In $\text{MEST}_{EM\&S}$, the density is reduced discretely at predefined epochs. In GraNet, the network evolution process consists of three steps: pruning, link removal, and link regrowth. The method first prunes the network to reduce the density, followed by removing and regrowing an equivalent number of links under the updated density. The density decrease in GraNet follows the same approach as Gradual Magnitude Pruning (GMP) [41], which adheres to a cubic function.

However, this density decay scheduler exhibits a sharp decline in the initial stages of training, which risks pruning a substantial fraction of weights before the model has sufficiently learned. To enable a smoother decay during the pruning stage, we propose a sigmoid-based gradual density decrease strategy, defined as:

$$s_t = s_i + (s_i - s_f)\left(\frac{1}{1 + e^{-k\left(t - \frac{t_f + t_0}{2}\right)}}\right), \tag{4}$$

where $t \in \{t_0, t_0 + \Delta t, \ldots, t_0 + n\Delta t\}$, $s_i$ is the initial sparsity, $s_f$ is the target sparsity, $t_0$ is the starting epoch of gradual pruning, $t_f$ is the end epoch of gradual pruning, and $\Delta t$ is the pruning frequency. $k$ controls the curvature of the decrease. We set $k=6$ for all the experiments in this article. This strategy ensures a smoother initial pruning phase, allowing the model to warm up and stabilize before undergoing significant pruning, thereby enhancing training stability and performance. We explain how to adjust the training FLOPs of sigmoid density decay to the same as cubic decay in Appendix I.

In addition to refining the decay function, we replace the weight magnitude criterion used in the original GMP and GraNet processes with relative importance (RI). This adjustment is motivated by prior work [32], which has shown that RI provides a significant performance advantage over weight magnitude, particularly when pruning models initialized with high density.

## 4 Experiments

### 4.1 Experimental Setup

We evaluate the performance of CHTs using MLPs for image classification tasks on the MNIST [16], Fashion MNIST [18], EMNIST [17], and CIFAR10 [42] datasets. To further validate our approach, we apply the sigmoid gradual density decay strategy to Transformers for machine translation tasks on the Multi30k en-de [20], IWSLT14 en-de [21], and WMT17 en-de [22] datasets. Additionally, we conduct language modeling experiments using the Open-WebText dataset [43] and evaluate zero-shot performance on the GLUE [44] and SuperGLUE [45] benchmark with LLaMA-60M, 130M, and 1B models [1]. For MLP training, we sparsify all layers except the final layer, as ultra-sparsity in the output layer may lead to disconnected neurons, and the connections in the final layer are relatively minor compared to the previous layers. For Transformers and LLaMA models, we apply dynamic sparse training (DST) to all linear layers, excluding the embedding and final generator layer. Detailed hyperparameter settings for each experiment are provided in Tables 4, 5, and 6. We also conduct a series of ablation and sensitivity tests on the components proposed in this article for CHTs and CHTss in Appendix H.

Table 1: Performance comparison of different sparsity dynamic sparse training methods on the CIFAR10 dataset trained on an MLP at 99% sparsity. The density decay of GMP, GraNet, and CHTss starts with a sparsity of 50%. ACC represents accuracy, and ANP denotes the active neuron percolation rate, indicating the final size of the network. The lowest anp rate and the best dynamic sparse training method are highlighted in bold, and performances surpassing the fully connected model are marked with "*". The results present a standard error taken over three seeds of the experiments.

| Method | ACC (%) | Comparison to FC | ANP |
|---|---|---|---|
| FC | $62.85 \pm 0.16$ | – | – |
| CHTs | $\mathbf{69.97 \pm 0.06}$* | **+11.33%** | **54%** |
| CHT | $59.10 \pm 0.06$ | -5.97% | 96% |
| RigL | $63.90 \pm 0.19$* | +1.67% | 59% |
| SET | $62.70 \pm 0.11$ | -0.24% | 100% |
| CHTss | $\mathbf{71.29 \pm 0.14}$* | **+13.43%** | 63% |
| GraNet | $69.31 \pm 0.17$* | +10.28% | **61%** |
| GMP | $65.11 \pm 0.11$* | +3.60% | 75% |

Table 2: Performance comparison on machine translation tasks of Multi30k, IWSLT, and WMT with varying final sparsity levels. The scores indicate BLEU scores, which is the higher the better. CHTs (GMP) represents CHTs with GMP's density decay strategy. Bold values denote the best performance among fixed sparsity DST methods or density decay DST methods. The performances that surpass the fully connected model are marked with "*". The density decay of GMP, GraNet, and CHTss starts with a sparsity of 50%. The scores are averaged over three seeds ± their standard error.

| Method | Multi30k | | IWSLT | | WMT | |
|---|---|---|---|---|---|---|
| | 95% | 90% | 95% | 90% | 95% | 90% |
| FC | 31.38 ± 0.38 | | 24.48 ± 0.30 | | 25.22 | |
| SET | 28.99 ± 0.28 | 29.73 ± 0.10 | 18.53 ± 0.05 | 20.13 ± 0.08 | 20.19 ± 0.12 | 21.52 ± 0.28 |
| RigL | 29.94 ± 0.27 | 30.26 ± 0.34 | 20.53 ± 0.21 | 21.52 ± 0.15 | 20.71 ± 0.21 | 22.22 ± 0.10 |
| CHT | 27.79 | 28.38 | 18.59 | 19.91 | 19.03 | 21.08 |
| CHTs | 28.94 ± 0.57 | 29.81 ± 0.37 | 21.15 ± 0.10 | 21.92 ± 0.17 | 20.94 ± 0.63 | 22.40 ± 0.06 |
| $MEST_{EM\&S}$ | 28.89± 0.26 | 30.04 ± 0.52 | 19.56 ± 0.10 | 21.05 ± 0.21 | 20.70 ± 0.07 | 22.22 ± 0.10 |
| GMP | 30.51 ± 0.82 | 30.49 ± 0.40 | 22.76 ± 0.82 | 22.82 ± 0.53 | 22.47 ± 0.10 | 23.37 ± 0.08 |
| GraNet | 31.31 ± 0.31 | 31.62 ± 0.48* | 22.53 ± 0.12 | 22.43 ± 0.09 | 22.51 ± 0.21 | 23.46 ± 0.09 |
| CHTss | **32.03 ± 0.29*** | **32.86 ± 0.16*** | **24.51 ± 0.02*** | **24.31 ± 0.04** | **23.73 ± 0.43** | **24.61 ± 0.14** |

**Baseline Methods.** We compare our method with the baseline approaches in the literature. We divide the dynamic sparse training (DST) methods into two categories: fixed sparsity DST and density decay DST. For the fixed sparsity DST, we compare CHTs with the SET [5], RigL [7], and CHT [9], and for the density decay DST methods, we compare CHTss with $MEST_{EM\&S}$ [8], GMP [41], and GraNet [40]. We explain the detailed implementations and the reason we split GMP as a type of DST method in Appendix G.

## 4.2 MLP for image classification

**Main results.** In the MLP evaluation, we aim to assess the fundamental capacity of DST methods to train the fully connected module, which is common across many ANNs. The sparse topological initialization of CHT and CHTs is CSTI [9] since the input bipartite layer can directly receive information from the input pixels. Table 7 displays the performance of DST methods compared to their fully connected counterparts across three basic datasets of MNIST, Fashion MNIST, and EMNIST. The DST methods are tested at 99% sparsity. As shown in Table 7, both of the two regrowth methods of CHTs outperform the other fixed sparsity DST methods. Notably, the path-based CH3-L3p outperforms the fully connected one in all the datasets. The node-based CH2-L3n also achieves comparable performance on these basic datasets. However, considering the running time of CH3-L3p is unacceptable, especially in large scale models, in the rest of the experiments of this article, we only use CH2-L3n as the representative method to regrow new links. Table 1 presents a comparison of fixed-sparsity dynamic sparse training (DST) methods against the fully connected (FC) baseline. Notably, CHTs outperforms all other DST methods and achieve an 11% improvement in accuracy over the fully connected model. In addition, we present the active neuron post-percolation rate (ANP) for each method in Table 7 and Table 1. It is evident that CHTs adaptively percolates the network more effectively while retaining performance.

## 4.3 Transformer on Machine Translation

We assess the Transformer's performance on a classic machine translation task across three datasets. We take the best performance of the model on the validation set and report the BLEU on the test set. Beam search, with a beam size of 2, is employed to optimize the evaluation process. In our evaluation, CHTs and CHTss configure the topology of each layer using the BRF model, employ a weight magnitude soft link removal technique, and regrow new links using CH2-L3n-soft. Additionally, we apply an adjusted network percolation technique to the Transformer, as detailed in Appendix F. The findings, presented in Table 2, demonstrate that 1) CHTs surpasses other fixed density DST methods on all the sparsity scenarios except for Multi30K. 2) Incorporating the sigmoid density decrease, CHTss outperforms even the fully connected ones with only 5% density on Multi30K and IWSLT.

Table 3: **Validation perplexity of different dynamic sparse training (DST) methods on Open-WebText using LLaMA-60M, LLaMA-130M, and LLaMA-1B across varying sparsity levels.** Bold values denote the best performance among fixed sparsity DST methods or density decay DST methods. Lower perplexity corresponds to better model performance. GMP, GraNet, and CHTss are run with an initial sparsity of $s_i = 0.5$. The test of CHT over LLaMA-1B is missing due to its excessive runtime. The performances that surpass the fully connected model are marked with "*".

| Method | LLaMA-60M | | | | LLaMA-130M | | | | LLaMA-1B |
|--------|-----|-----|-----|-----|-----|-----|-----|-----|-----|
| | 70% | 80% | 90% | 95% | 70% | 80% | 90% | 95% | 70% |
| FC | 26.56 | | | | 19.27 | | | | 14.62 |
| SET | 31.77 | 30.69 | 35.26 | 39.70 | 20.82 | 22.02 | 24.73 | 28.37 | 16.37 |
| RigL | 39.96 | 41.33 | 45.34 | 51.49 | 25.85 | 66.35 | 37.18 | 49.39 | 149.17 |
| CHT | 31.02 | 32.99 | 35.01 | 41.87 | 21.02 | 22.82 | 26.27 | 30.01 | – |
| CHTs | **28.12** | **29.84** | **33.03** | **36.47** | **20.10** | **21.33** | **23.71** | **26.45** | **14.53*** |
| MEST | 28.26 | 29.94 | 33.60 | 37.87 | 21.32 | 22.21 | 24.98 | 27.96 | 60.36 |
| GMP | 29.22 | 30.59 | 33.68 | 39.00 | 20.49 | 22.28 | 23.61 | 27.16 | 31.76 |
| GraNet | 30.55 | 31.51 | 33.76 | 39.98 | 22.84 | 29.03 | 26.81 | 61.31 | 79.44 |
| CHTss | **27.62** | **29.00** | **31.42** | **35.10** | **19.85** | **20.70** | **22.51** | **25.07** | **15.41** |

## 4.4 Natural Language Generation

**Language modeling.** We utilize the LLaMA model family [1] across 60M, 130M, and 1B architecture as the baseline for our language generation experiments. We follow the experiment setup from [46] detailed in Table 6. To ensure that the FLOPs are the same for all the density decrease methods, for CHTss, we implement a half-step strategy for the density decay steps to make sure the sparsity across the training steps is the same as GMP and GraNet.

Table 3 shows the validation perplexity results of each algorithm across different density levels on LLaMA-60M and LLaMA-130M. CHTs stably outperforms SET, CHT, and RigL, while CHTss are constantly better than GraNet and GMP. At 70% sparsity, CHTss is already able to perform comparably to the fully connected model. It is important to note that RigL and GraNet exhibit subpar performance in this evaluation due to the use of bfloat16 precision in this configuration. This lower precision adversely impacts gradient accuracy, particularly in the early stages of training. Since both RigL and GraNet rely on gradient information to regrow new links, the imprecise gradients lead to regrowing the wrong links, thereby hindering their performance. To further validate this observation, we conduct additional experiments under FP32 precision, presented in Table 10. The results confirm that RigL and GraNet perform significantly better under high-precision training, although they still fall behind CHTs and CHTss. Importantly, industry trends are increasingly moving toward low-precision training and inference to improve efficiency [47, 48]. In this context, CHTs and CHTss demonstrate greater robustness compared to RigL and GraNet, making them better aligned with practical deployment needs under reduced-precision settings.

## 5 Conclusion and Discussion

We advance current knowledge in brain-inspired dynamic sparse training, proposing the Cannistraci-Hebb Training soft rule with sigmoid gradual density decay (CHTss). First, we introduce a matrix multiplication mathematical formula for GPU-friendly approximation of the CH link predictor. This significantly reduces the computational complexity of CHT and speeds up the running time, enabling the implementation of CHTs in large-scale models. Second, we propose a Cannistraci-Hebb training soft rule (CHTs), which innovatively utilizes a soft sampling rule for both removal and regrowth links, striking a balance for epitopological exploration and exploitation. Third, we propose the Bipartite Receptive Field (BRF) model to initialize the sparse network topology in a brain-inspired manner, enabling the network to preferentially capture spatially closer features. Finally, in transformer-based models, we integrate CHTs with a sigmoid gradual density decay strategy (CHTss). Empirically, CHTs demonstrate a remarkable ability to achieve ultra-sparse configurations—up to 99% sparsity in MLPs for image classification—surpassing fully connected networks. Notably, the regrow process under CHTs does not rely on gradients. With the sigmoid gradual density decay, CHTss surpasses

the fully connected Transformer using only 5% density and achieves comparable language modeling performance. Moreover, CHTs and CHTss achieve language modeling performance comparable to the fully connected LLaMA-60M and LLaMA-130M models, while CHTs even surpasses the fully connected LLaMA-1B counterpart. We describe the limitations of this study and future works in Appendix A.

## Acknowledgements

This work was supported by the Zhou Yahui Chair Professorship award of Tsinghua University (to CVC), the National High-Level Talent Program of the Ministry of Science and Technology of China (grant number 20241710001, to CVC).

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

Table 4: **Hyperparameters of MLP on Image Classification Tasks.**

| Hyper-parameter | MLP |
|---|---|
| Hidden Dimension | 1568 (3072 for CIFAR10) |
| # Hidden layers | 3 |
| Batch Size | 32 |
| Training Epochs | 100 |
| LR Decay Method | Linear |
| Start Learning Rate | 0.025 |
| End Learning Rate | $2.5e^{-4}$ |
| $\zeta$ (fraction of removal) | 0.3 |
| Update Interval (for DST) | 1 epoch |
| Momentum | 0.9 |
| Weight decay | $5e^{-4}$ |

Table 5: **Hyperparameters of Transformer on Machine Translation Tasks.** `inoam` refers to a learning rate scheduler that incorporates iterative warm-up phases, specifically designed for dynamic sparse training (DST) methods. The purpose is to allow newly regrown connections to accumulate momentum, preventing potential harm to the training process. For the fully connected (FC) baseline, only the standard `noam` scheduler is used.

| Hyper-parameter | Multi30k | IWSLT14 | WMT17 |
|---|---|---|---|
| Embedding Dimension | 512 | 512 | 512 |
| Feed-forward Dimension | 1024 | 2048 | 2048 |
| Batch Size | 1024 tokens | 10240 tokens | 12000 tokens |
| Training Steps | 5000 | 20000 | 80000 |
| Dropout | 0.1 | 0.1 | 0.1 |
| Attention Dropout | 0.1 | 0.1 | 0.1 |
| Max Gradient Norm | 0 | 0 | 0 |
| Warmup Steps | 1000 | 6000 | 8000 |
| Learning rate Decay Method | inoam | inoam | inoam |
| Iterative warmup steps | 20 | 20 | 20 |
| Label Smoothing | 0.1 | 0.1 | 0.1 |
| Layer Number | 6 | 6 | 6 |
| Head Number | 8 | 8 | 8 |
| Learning Rate | 0.25 | 2 | 2 |
| $\zeta$ (fraction of removal) | 0.3 | 0.3 | 0.3 |
| Update Interval (for DST) | 100 steps | 100 steps | 100 steps |

# A   Limitations and Future Work

A potential limitation of this work is that the hardware required to accelerate sparse training with unstructured sparsity has not yet become widely adopted. Consequently, this article does not present a direct comparison of training speeds with those of fully connected networks. However, several leading companies [49, 50] have already released devices that support unstructured sparsity in training.

For future work, we aim to develop methods for automatically determining the temperature for soft sampling at each epoch, guided by the topological features of each layer. This could enable each layer to learn its specific topological rules autonomously. Additionally, we plan to test CHTs and CHTss in larger LLMs such as LLaMA-7b to evaluate the performance in scenarios with denser layers.

# B   Broader Impact

In this work, we introduce a novel methodology for dynamic sparse training aimed at enhancing the efficiency of AI model training. This advancement holds potential societal benefits by increasing interest in more efficient AI practices. However, the widespread availability of advanced artificial

Table 6: **Hyperparameters of LLaMA-60M, LLaMA-130M, and LLaMA-1B on OpenWebText.** `inoam` refers to a learning rate scheduler that incorporates iterative warm-up phases, specifically designed for dynamic sparse training (DST) methods. The purpose is to allow newly regrown connections to accumulate momentum, preventing potential harm to the training process. For the fully connected (FC) baseline, only the standard `noam` scheduler is used.

| Hyper-parameter | LLaMA-60M | LLaMA-130M | LLaMA-1B |
|---|---|---|---|
| Embedding Dimension | 512 | 768 | 2048 |
| Feed-forward Dimension | 1376 | 2048 | 5461 |
| Global Batch Size | 512 | 512 | 512 |
| Sequence Length | 256 | 256 | 256 |
| Training Steps | 10000 | 30000 | 100000 |
| Learning Rate | 3e-3 (1e-3 for FC) | 3e-3 (1e-3 for FC) | 3e-3 (4e-4 for FC) |
| Warmup Steps | 1000 | 10000 | 10000 |
| Learning rate Decay Method | inoam | inoam | inoam |
| Iterative warmup steps | 20 | 20 | 20 |
| Optimizer | Adam | Adam | Adam |
| Layer Number | 8 | 12 | 24 |
| Head Number | 8 | 12 | 32 |
| $\zeta$ (fraction of removal) | 0.1 | 0.1 | 0.1 |
| Update Interval (for DST) | 100 steps | 100 steps | 100 steps |

Table 7: Performance comparison of different dynamic sparse training methods on MNIST, Fashion MNIST (FMNIST), and EMNIST datasets trained on MLP at 99% sparsity. ACC represents accuracy, and ANP denotes the active neuron percolation rate, indicating the final size of the network. Accuracies present a standard error taken over three seeds. The best dynamic sparse training method for each dataset is highlighted in bold, and the performances that surpass the fully connected model are marked with "*".

| Method | MNIST | | FMNIST | | EMNIST | |
|---|---|---|---|---|---|---|
| | ACC (%) | ANP | ACC (%) | ANP | ACC (%) | ANP |
| FC | $98.78 \pm 0.02$ | – | $90.88 \pm 0.02$ | – | $87.13 \pm 0.04$ | – |
| CHTs$^p$ | **$98.81 \pm 0.04$*** | 20% | **$90.93 \pm 0.03$*** | 89% | $87.61 \pm 0.07$* | 24% |
| CHTs$^n$ | $98.76 \pm 0.05$ | 27% | $90.67 \pm 0.05$ | 73% | **$87.82 \pm 0.04$*** | 28% |
| CHT | $98.48 \pm 0.04$ | 29% | $88.70 \pm 0.07$ | 30% | $86.35 \pm 0.08$ | 21% |
| RigL | $98.61 \pm 0.01$ | 29% | $89.91 \pm 0.07$ | 55% | $86.94 \pm 0.08$ | 28% |
| SET | $98.14 \pm 0.02$ | 100% | $89.00 \pm 0.09$ | 100% | $86.31 \pm 0.08$ | 100% |
| CHTss$^n$ | **$98.83 \pm 0.02$*** | 32% | **$90.81 \pm 0.11$** | 40% | **$87.52 \pm 0.04$*** | 35 % |
| GraNet | $98.81 \pm 0.00$* | 35% | $89.98 \pm 0.06$ | 53% | $86.94 \pm 0.03$ | 45% |
| GMP | $98.62 \pm 0.03$ | 58 % | $90.29 \pm 0.19$ | 69% | $86.93 \pm 0.09$ | 75 % |

[p] Refers to the regrowth method CH3_L3p.
[n] Refers to the regrowth method CH2_L3n.

neural networks, particularly large language models (LLMs), also presents risks of misuse. It is essential to carefully consider and manage these factors to maximize benefits and minimize risks.

## C  Cannistraci-Hebb epitopological rationale

The original CHT framework leverages the Cannistraci-Hebb link predictor on Length 3 paths (CH3-L3p) metric for link regrowth. Given two seed nodes $u$ and $v$ in a network, this metric assigns a score

$$\textbf{CH3-L3p}(u,v) = \sum_{z_1,z_2 \in L3} \frac{1}{\sqrt{de^*_{z_1} \cdot de^*_{z_2}}} \qquad (5)$$

Table 8: Perplexity (PPL) results across different sparsities (0.7, 0.8, 0.9, 0.95) for CHTs and CHTss under different regrowth strategies (Fixed and Uniform) and $r$ settings on LLaMA60M.

| | | **Fixed** | | | | **Uniform** | | | |
|---|---|---|---|---|---|---|---|---|---|
| | Sparsity | $r=0.0$ | $r=0.1$ | $r=0.2$ | $r=0.3$ | $r=0.0$ | $r=0.1$ | $r=0.2$ | $r=0.3$ |
| **CHTs** | 70% | 28.16 | 28.39 | 28.25 | 28.32 | 30.11 | **28.12** | 28.43 | 28.56 |
| | 80% | 30.22 | **29.84** | 30.04 | 30.03 | 30.19 | 29.86 | 30.23 | 30.06 |
| | 90% | 33.32 | 33.37 | **33.03** | 33.77 | 33.45 | 33.36 | 33.88 | 33.72 |
| | 95% | 37.29 | 37.51 | 37.24 | 37.46 | 37.23 | **36.47** | 37.33 | 37.67 |
| **CHTss** | 70% | **27.62** | 30.05 | 27.82 | 28.43 | **27.62** | 27.74 | 27.74 | 27.68 |
| | 80% | **29.00** | **29.00** | 29.66 | 32.91 | 29.49 | 29.69 | 29.09 | 29.24 |
| | 90% | 31.51 | 31.67 | 31.65 | 31.59 | 31.66 | 32.61 | 31.68 | **31.42** |
| | 95% | 38.66 | 35.31 | 36.24 | 37.50 | 42.20 | 37.40 | 35.36 | **35.10** |

Table 9: Perplexity (PPL) results across different sparsities (0.7, 0.8, 0.9, 0.95) for CHTs and CHTss under different regrowth strategies (Fixed and Uniform) and $r$ settings on LLaMA-130M.

| | | **Fixed** | | | | **Uniform** | | | |
|---|---|---|---|---|---|---|---|---|---|
| | Sparsity | $r=0.0$ | $r=0.1$ | $r=0.2$ | $r=0.3$ | $r=0.0$ | $r=0.1$ | $r=0.2$ | $r=0.3$ |
| **CHTs** | 70% | 20.24 | 20.16 | **20.10** | 20.25 | 20.62 | 20.18 | 20.15 | 20.20 |
| | 80% | **21.33** | 21.37 | 21.36 | 21.48 | 21.34 | 21.40 | 21.40 | 22.49 |
| | 90% | 23.72 | 23.76 | 23.76 | 23.94 | 23.74 | 23.73 | **23.71** | 24.99 |
| | 95% | 28.05 | **26.45** | 26.90 | 26.91 | 26.78 | 27.97 | 29.05 | 27.10 |
| **CHTss** | 70% | 20.63 | 19.88 | 19.93 | **19.85** | 21.43 | 19.90 | 20.93 | 19.94 |
| | 80% | 20.71 | 22.60 | 20.86 | **20.70** | 20.73 | 20.74 | 20.72 | 20.82 |
| | 90% | 22.58 | 22.72 | 22.61 | **22.51** | 22.53 | 22.59 | 22.60 | 23.12 |
| | 95% | 25.28 | 25.12 | 25.20 | 25.12 | **25.07** | 25.15 | 25.23 | 25.12 |

Here, $u$ and $v$ denote the seed nodes, while $z_1$ and $z_2$ are common neighbors on the L3 path [30], a walk of three consecutive links that connects $u$ to $v$ via those two intermediate nodes. The term $de_i^*$ represents the number of external local community links (eLCL) of node $i$, with a default increment of 1 to prevent division by zero. Path-based link prediction has demonstrated its effectiveness on both real-world networks [30] and artificial neural networks [9]. However, this method incurs a high computational cost due to the need to compute and store all length-three paths, resulting in a time complexity of $O(N \cdot d^3)$, where $N$ is the number of nodes and $d$ is the network's average degree. This complexity is prohibitive for large models with numerous nodes and higher-density layers. To

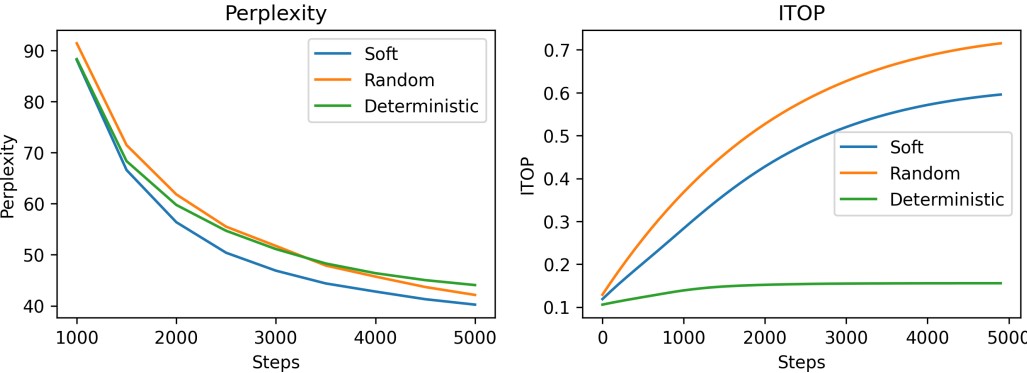

Figure 3: Comparison of link regrowth strategies in CHTs using a LLaMA-60M model trained on OpenWebText for 5000 steps. The left plot shows validation perplexity (lower is better), while the right plot reports the in-time over-parameterization (ITOP) rate, which measures the cumulative proportion of links activated during training. Results are presented for three strategies: Soft, Random, and Deterministic regrowth.

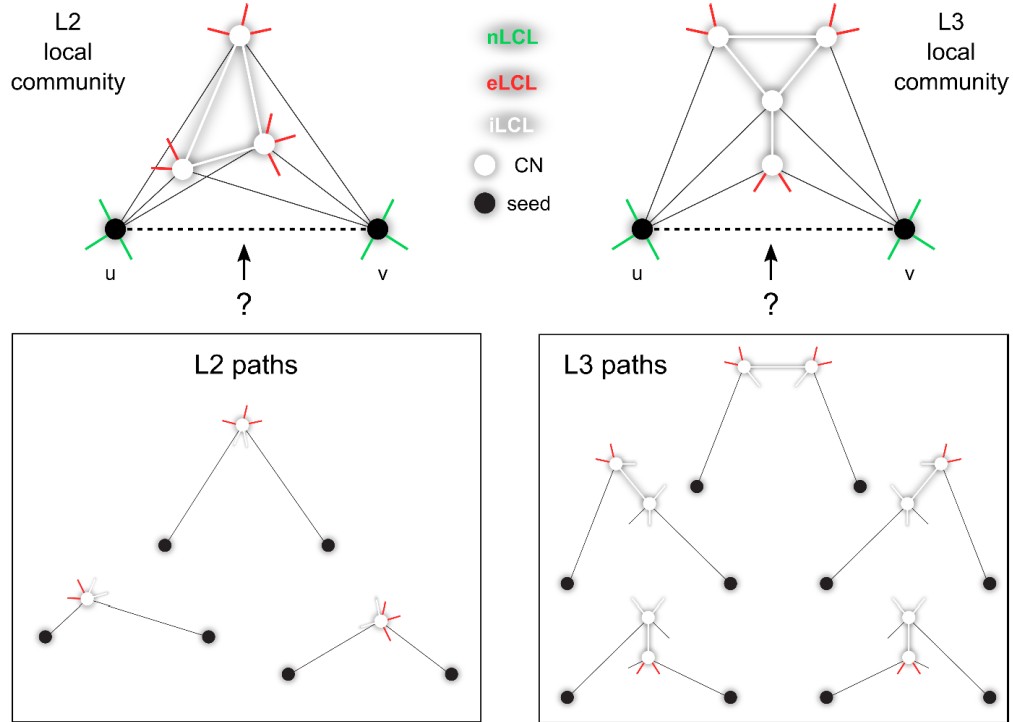

Figure 4: **Cannistraci-Hebb epitopological rationale.** [30] The figure illustrates an explanatory example of topological link prediction using the Cannistraci-Hebb epitopological rationale based on either L2 or L3 paths. The two black nodes represent the seed nodes whose unobserved interaction is to be assigned a likelihood score. White nodes denote the common neighbours (CNs) of the seed nodes at either L2 or L3 distance. Together, the set of CNs and the internal local community links (iLCL) constitute the local community. Different link types are color-coded: green for nLCLs, red for external local community links (eLCLs), and white for iLCLs. The L2 (path length 2) and L3 (path length 3) paths associated with the illustrated communities are highlighted. Notably, in artificial neural networks (ANNs), linear layers correspond to bipartite networks, which inherently support only L3 path predictions, as shown in Figure 1.

address this issue, we introduce a more efficient, node-based paradigm that eliminates the reliance on length-three paths between seed nodes. Instead, this approach focuses on the common neighbors of seed nodes. The node-based version of CH3-L3p, denoted as CH2-L3n, is defined as follows:

$$\textbf{CH2-L3n}(u, v) = \sum_{z \in L3} \frac{di_z^*}{de_z^*} \tag{6}$$

Here, $u$ and $v$ denote the seed nodes, while $z$ is the common neighbor on the L3 path [30], a walk of three consecutive links that connects $u$ to $v$ via two of those intermediate nodes. The terms $di_z^*\, de_z^*$ represent the number of internal local community links (iLCLs) and external local community links (eLCLs) of node $i$, with a default increment of 1 to prevent division by zero. Internal local community links (iLCLs) are those that connect nodes belonging to the same local community. Contrarily, external local community links (eLCLs) connect nodes belonging to different communities. Figure 4 gives a visual representation of L2 and L3 paths between two seed nodes $u$ and $v$, defining their local community.

## D   Sparse topological initialization

**Correlated sparse topological initialization.**   Correlated Sparse Topological Initialization (CSTI) is a physics-informed topological initialization. CSTI generates the adjacency matrix by computing the Pearson correlation between each input feature across the calibration dataset and then selects

the predetermined number of links, calculated based on the desired sparsity level, as the existing connections. CSTI performs remarkably better when the layer can directly receive input information. However, for layers that cannot receive inputs directly, it cannot capture the correlations from the start since the model is initialized randomly, as in the case of the Transformer. Therefore, in this article, we aim to address this issue by investigating different network models to initialize the topology, to improve the performance for cases where CSTI cannot be directly applied.

**Bipartite scale-free model.**    In artificial neural networks (ANNs), fully connected networks are inherently bipartite. This article explores initializing bipartite networks using models from network science. The Bipartite Scale-Free (BSF) [9] network model extends the concept of scale-freeness to bipartite structures, making them suitable for dynamic sparse training. Initially, the BSF model generates a monopartite Barabási-Albert (BA) model [37], a well-established method for creating scale-free networks in which the degree distribution follows a power law ($\gamma$=2.76 in Figure 5). Following the creation of the BA model, the BSF approach removes any connections between nodes of the same type (neuron in the same layer) and rewires these connections to nodes of the opposite type (neuron in the opposite layer). This rewiring is done while maintaining the degree of each node constant to preserve the power-law exponent $\gamma$.

**Bipartite small-world model.**    The Bipartite Small-World (BSW) network model [9] is designed to incorporate small-world properties and a high clustering coefficient into bipartite networks. Initially, the model constructs a regular ring lattice and assigns two distinct types of nodes to it. Each node is connected by an equal number of links to the nearest nodes of the opposite type, fostering high clustering but lacking the small-world property. Similar to the Watts-Strogatz model (WS) [36], the BSW model introduces a rewiring parameter, $\beta$, which represents the percentage of links randomly removed and then rewired within the network. At $\beta = 1$, the model transitions into an **Erdős-Rényi model** [51], exhibiting small-world properties but without a high clustering coefficient, which is popular as the topological initialization of the other dynamic sparse training methods [5, 7, 8].

**Bipartite receptive field model.**    The Bipartite Receptive Field (BRF) model is a random network generation technique designed to mimic the receptive field phenomenon in the brain networks. The process involves adding links to the adjacency matrix of the bipartite network, with the connectivity structured around the main diagonal according to a parameter $r \in [0, 1]$. A low value of $r$ results in links that are primarily clustered around the diagonal, while a higher value of $r$ leads to a more random connectivity pattern. Specifically, a bipartite adjacency matrix with links near the diagonal indicates that adjacent nodes from the two layers are linked, whereas links far from the diagonal correspond to more distant node pairs. Mathematically, consider an $N \times M$ bipartite adjacency matrix $M_{i,j\ i=1,...,M,j=1,...,N}$, where $M$ represents the input size and $N$ represents the output size. Each entry of the matrix $m_{i,j}$ is set to 1 if input node $i$ is connected to output node $j$, and 0 otherwise. A scoring function $S_{i,j}$ is assigned to each connection in the adjacency matrix based on its distance to the main diagonal. This score is given by:

$$S_{i,j} = d_{ij}^{\frac{1-r}{r}}, \tag{7}$$

where

$$d_{ij} = min\{|i - j|, |(i - M) - j|, |i - (j - N)|\} \tag{8}$$

is the distance between the input and output neurons. Therefore, $S_{i,j}$ represents the distance from the diagonal, raised to the power of $\frac{1-r}{r}$. The parameter $r$ controls how structured or random the adjacency matrix is. As $r \to 0$, the scoring function becomes more deterministic, with high scores assigned to entries near the diagonal and low scores to entries farther away. Conversely, as $r \to 1$, all scores $S_{i,j}$ become more uniform, leading to a more random, less structured adjacency matrix. The next step is to determine the degree distribution for the output nodes. This can either be fixed, assigning the same degree to all output nodes, or uniform, where the degrees are randomly sampled from a uniform distribution. Hence, we propose two variations of the BRF model: the Bipartite Receptive Field with fixed sampling (BRFf), in which the degrees of output nodes are fixed, and the Bipartite Receptive Field with uniform sampling (BRFu), where the degrees of the output nodes follow a uniform distribution. This represents an additional enhancement to the WS scheme, which offers no way to control how connections are allocated among the output nodes. In conclusion, to run the BRF model, the user should input an output degree distribution and a spatial dependent distance randomness.

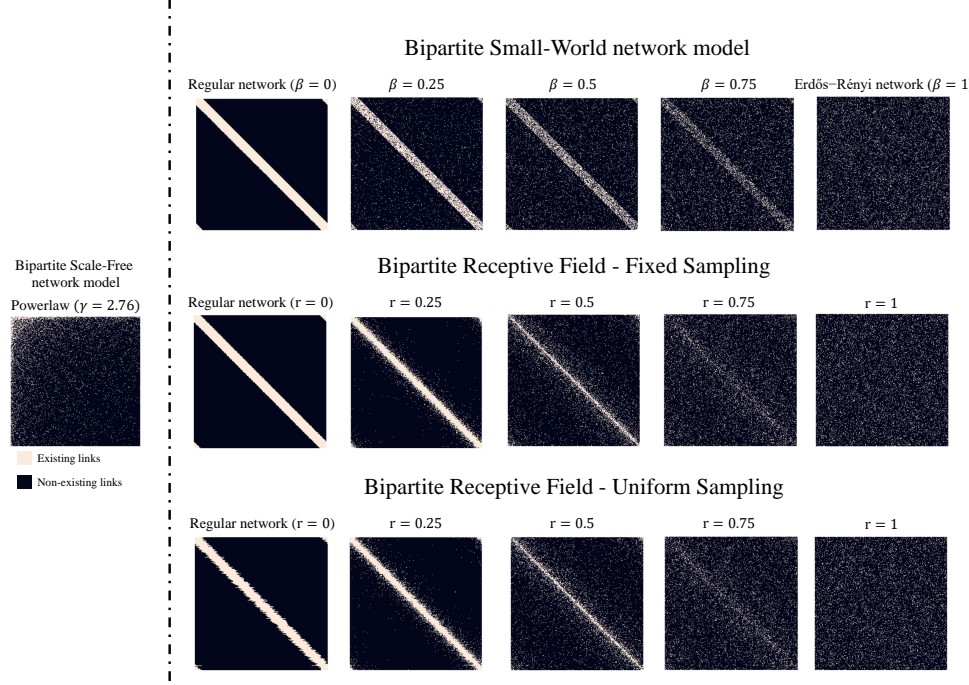

Figure 5: **The adjacency matrix** of the Bipartite Scale-Free (BSF) network model compared to those of the Bipartite Small-World (BSW) network, the Bipartite Receptive Field with fixed sampling ($BRF_f$), and the Bipartite Receptive field with uniform sampling ($BRF_u$) as parameters $\beta$ and $r$ vary between 0 and 1. a) The BSF model inherently forms a scale-free network characterized by a power-law distribution with $\gamma = 2.76$. b) As $\beta$ changes from 0 to 1, the network exhibits reduced clustering. It is important to note that when $\beta = 0$, the BSW model does not qualify as a small-world network. c) As $r$ increases towards 1, the adjacency matrix becomes more random, while sampling the output neurons' degrees from a fixed or uniform distribution.

## E   Equal Partition and Neuron Resorting to enhance bipartite scale-free network initialization

As indicated in SET and CHT [5, 9], trained sparse models typically converge to a scale-free network. This suggests that initiating the network with a scale-free structure might initially enhance performance. However, starting directly with a Bipartite Scale-Free model (BSF, power-law exponent $\gamma = 2.76$) does not yield effective results. Upon deeper examination, two potential reasons emerge:

- The BSF model generates hub nodes randomly. However, this random assignment of hub nodes to less significant inputs leads to a less effective initialization, which is particularly detrimental in CHT, which merely utilizes the topology information to regrow new links.

- As demonstrated in CHT, in the final network, the hub nodes of one layer's output should correspond to the input layer of the subsequent layer, which means the hub nodes should have a high degree on both sides of the layer. However, the BSF model's random selection disrupts this correspondence, significantly reducing the number of Credit Assignment Paths (CAP) [9] in the model. CAP is defined as the chain of transformation from input to output, which counts the number of links that go through the hub nodes in the middle layers.

To address these issues, we propose two solutions:

- Equal Partitioning of the First Layer: We begin by generating a BSF model, then rewire the connections from the input layer to the first hidden layer. While keeping the out-degrees of the output neurons fixed, we randomly sample new connections to the input neurons until each of the input neurons' in-degrees reaches the input layer's average in-degree. This

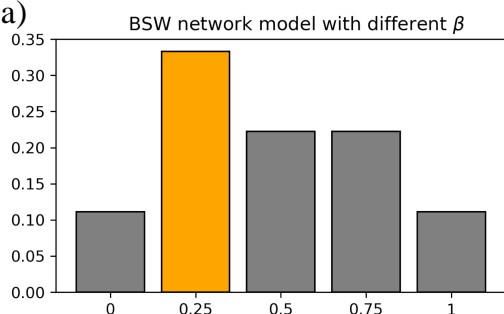
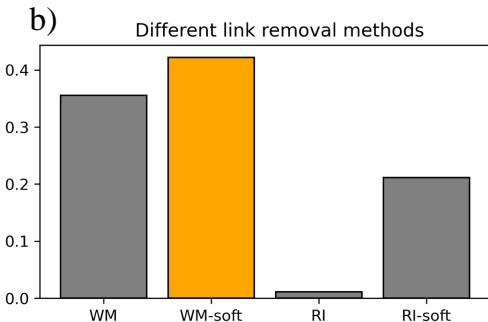

Figure 6: **The ablation test** of the $\beta$ of the bipartite small world (BSW) model and the removal methods in CHTs. a) evaluates the influence of the rewiring rate $\beta$ on the model performance when initialized with the BSW model. b) assesses the influence of link removal selecting from the weight magnitude (WM), weight magnitude soft (WM-soft), relative importance (RI), and relative importance soft (RI-soft). We utilize the win rate of the compared factors under the same setting across each realization of 3 seeds for all experiment combinations on MLP. The factor with the highest win rate is highlighted in orange.

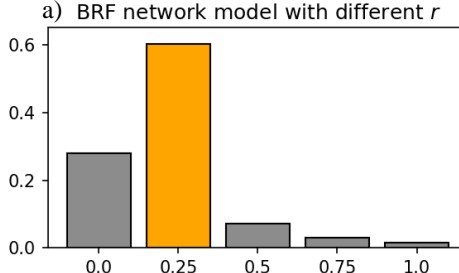
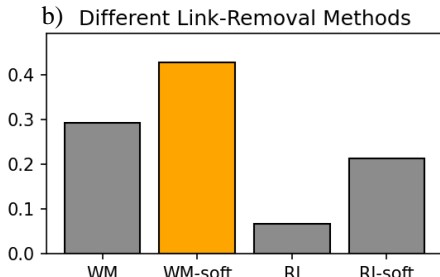

Figure 7: **The ablation test of** the parameter $r$ in the bipartite receptive field (BRF) model and the removal methods in CHTs using the BRF initialization technique. a) evaluates the influence of the parameter $r$ on the model performance when initialized with the BRF model. b) assesses the influence of link removal in the CHTs model with BRF initialization. We utilize the win rate of the compared factors under the same setting across each realization of 3 seeds for all experiment combinations on MLP. The factor with the highest win rate is highlighted in orange.

> approach ensures all input neurons are assigned equal importance while maintaining the power-law degree distribution of output neurons.
>
> • Resorting Middle Layer Neurons: Given the mismatch in hub nodes between consecutive layers, we suggest permuting the neurons between the output of one layer and the input of the next, based on their degree. A higher degree in an output neuron increases the likelihood of connecting to a high-degree input neuron in the subsequent layer, thus enhancing the number of CAPs.

As illustrated in Figure 8, while the two techniques enhance the performance of the BSF initialization, they remain inferior to the BSW initialization. As noted in the main text, achieving scale-freeness is more effective when the model is allowed to learn and adapt dynamically rather than being directly initialized as a predefined structure.

## F   Network percolation and extension to Transformer.

We have adapted network percolation [52, 9] to suit the architecture of the Transformer after link removal. The core idea is to identify inactive neurons, which are characterized by having no

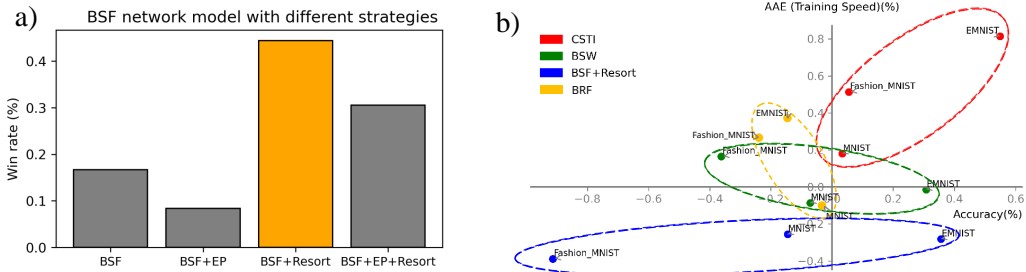

Figure 8: **The Performance** of the bipartite scale-free model and two enhanced techniques. a) shows the win rate of the Bipartite Scale-Free network model (BSF) with the different techniques. *EP* stands for equal partition of the first layer, and *Resort* refers to reordering the neurons based on their degree. b) assesses the comparison between Correlated Sparse Topological Initialization (CSTI), the Bipartite Scale-Free (BSF) model with the best solution from a), and the Bipartite Small-World (BSW) model with $\beta = 0.25$.

connections on either one or both sides within a layer of neurons. Such neurons disrupt the flow of information during forward propagation or backpropagation. In addition, Layer-wise computation of the CH link prediction score further implies that neurons without connections on one side are unlikely to form connections in the future. Therefore, network percolation becomes essential to optimize the use of remaining links.

As shown in Figure 1, network percolation encompasses two primary processes: c1) inactive neuron removal to remove the neurons that lack connections on one or both sides; c2) incomplete path adjustment to remove the incomplete paths where links connect to the inactive neurons after c1). Typically applied in simpler continuous layers like those in an MLP, network percolation requires modification for more complex structures. For example, within the Transformer's self-attention module, the outputs of the query and key layers undergo a dot product operation. It necessitates percolation in these layers to examine the activity of the neurons in both output layers at the same position. Similar interventions are necessary in the up_proj and gate_proj layers of the MLP module in the LLaMA model family [1, 53].

## G    Baseline Methods

### G.1    Fixed Density Dynamic Sparse Training Methods

**SET**    [5]: Removes connections based on weight magnitude and randomly regrows new links.

**RigL**    [7]: Removes connections based on weight magnitude and regrows links using gradient information, gradually reducing the proportion of updated connections over time.

**CHT**    [9]: A state-of-the-art (SOTA) gradient-free method that removes links with weight magnitude and regrows links based on CH3-L3 scores. CHT is often applied with early stopping to mitigate its computational complexity when working with large models.

### G.2    Gradual Density Decrease Dynamic Sparse Training Methods

**GMP**    [54, 41]: Prunes the network with weight magnitude and gradually decreases the density based on Equation 10. Although originally a pruning method, GMP is treated as a dynamic sparse training method in their implementation [41], as it stores historical weights and allows pruned weights to reappear during training, since, during training, the pruning threshold might change.

**MEST**$_{EM\&S}$    [8]: Implements a two-stage density decrease strategy as described in the original work. It removes links based on the combination of weight magnitude and 0.01*gradient and regrows new links randomly.

**GraNet** [40]: Gradually decreases density using Equation 10. Similar to RigL, GraNet removes links based on the weight magnitude and regrows new links with the gradient of the existing links.

Table 10: Float32 Precision Comparison on LLaMA-130M. Bold values denote the best performance among DST methods. Lower perplexity corresponds to better model performance. $s_i$ represents the initial sparsity for DST methods employing a density decay strategy.

| Method | Sparsity | |
| --- | --- | --- |
| | 70% | 80% |
| FC | 17.07 | |
| RigL | 18.34 | 19.64 |
| CHTs | 17.99 | 19.25 |
| GraNet ($s_i = 0.5$) | 17.92 | 18.79 |
| CHTss ($s_i = 0.5$) | **17.76** | **18.69** |

## H  Ablation and Sensitivity Tests

**An overall ablation test**  To fully assess each component's effectiveness, we conduct several ablation and sensitivity tests that help us understand how to select a sparse topological initialization and identify the best link removal and regrowth methods. We first made a global test for all the components in Table 11, which shows the effectiveness of each element introduced by this article. The node-based and path-based link regrowth methods have comparable performance, but the node-based versions are much faster.

**Sparse topological initialization.**  For sparse topological initialization, we compare BRF, BSW, BSF, and CSTI [9] across three image classification datasets, as shown in Figure 8b. The results indicate that when the inputs can directly access task-relevant information, CSTI consistently achieves the best performance. In general, BRF and BSW perform similarly under these conditions, but outperform the BSF initialization.

To further validate our findings, we evaluate BRF and BSW network initializations on machine translation tasks using Transformer models. Figure 9 and Figure 10 present the performance comparisons between BSW and BRF on the Multi30k and IWSLT datasets, while Figure 11 shows the win-rate analysis. These comparisons demonstrate that BRF consistently outperforms BSW across most cases. Additionally, Figure 7a analyzes the impact of the receptive field range $r$ on BRF initialization for MNIST, Fashion MNIST, and EMNIST tasks using MLPs, with results indicating that $r = 0.25$ yields the best performance.

Building on this prior knowledge, we further evaluate BRF on LLaMA-60M and LLaMA-130M models, testing $r$ values in the range $[0, 0.3]$ and comparing two different degree distributions. The results, shown in Table 8 and Table 9, indicate that on LLaMA models, the choice of $r$ and distribution has limited impact. While $r = 0.1$ wins slightly more often, the improvements remain marginal. Finally, Table 3 reports the best performance combinations of $r$ and degree distributions derived from these evaluations.

**Link removal.**  We first conduct a simple evaluation of the link removal methods introduced in this article when changing the $\alpha$ and $\delta$ inside Figure 1b2) on Figure 6b) and Figure 7b). The removal methods are selected from Weight Magnitude (WM), Weight Magnitude soft (WMs), Relative Importance (RI), and Relative Importance soft (RIs). For WM we fix the hyperparameters $\alpha = 1$ and $\delta = 1$; for RI we fix the hyperparameters $\alpha = 0$ and $\delta = 0.5$; for WMs we fix $\alpha = 1$ and we let $\delta$ increase linearly from 0.5 to 0.9; for RIs we fix $\alpha = 0$ and let $\delta$ increase linearly from 0.5 to 0.9. From the results, it can be observed that WMs performs the best in most cases. We compare these methods with those in [55] in Table 12 on two machine translation tasks. The results indicate that using WMs as a link removal method generally outperforms the alternatives.

We also evaluate how to define the softness in WMs. During sampling, we have a hyperparameter to decide the temperature of the scores that convert to the probability of being removed. We perform a test using a linear decay solution, since, generally, the weights in the model become more reliable as

Table 11: Ablation results of Transformer on Multi30K and IWSLT datasets at 90% sparsity. The scores indicate BLEU scores, the higher the better. Bold values denote the best performance among DST methods.

| Variant | Multi30K (90% sparsity) | IWSLT (90% sparsity) |
|---|---|---|
| a. CHT | 28.38 | 19.91 |
| b. CHTss without node-based implementation | 32.68 (2.42 hours) | **24.82** (18 hours) |
| c. CHTss without soft sampling | 28.92 | 21.88 |
| d. CHTss without sigmoid decay (= CHTs) | 30.35 | 21.60 |
| e. CHTss (full model) | **32.79** (0.25 hours) | 24.57 (1.5 hours) |

Table 12: Performance comparison of CHTs and CHTss at 90% sparsity across different removal methods. The tested dataset is Multi30K, and the reported metric is BLEU, which is the higher the better.

| Remove Method | CHTs | CHTss |
|---|---|---|
| set | 28.82 | 25.76 |
| wm | 28.17 | 31.15 |
| wm_soft | **30.35** | **32.79** |
| ri | 28.91 | 32.20 |
| ri_soft | 27.86 | 31.86 |
| MEST | 28.70 | 32.07 |
| snip | 28.23 | 31.66 |
| sensitivity | 29.02 | 29.73 |
| Rsensitivity | 28.18 | 30.67 |

training progresses. Figure 12 shows the variation in BLEU scores as we change the starting and ending values of the $\delta$ parameter in the soft weight magnitude removal method on transformer models. Recalling that we define the temperature by $T = \frac{1}{1-\delta}$, we observe that for a simple benchmark like Multi30k, a high starting temperature produces better performance. This is motivated by the fact that loss decreases very fast through epochs, meaning that weights are learned quickly, and we can deterministically remove weights with high reliability. In more complex datasets, like IWSLT, low starting temperatures are preferred. This is because during the early stages of training, weights are learned slowly, meaning that a deterministic removal can be less reliable. To be more consistent, we select a start $\delta = 0.5$ and end $\delta = 0.9$ for all the tasks in the main article.

**Density Decay Strategy.** To further demonstrate the benefit of the density decay strategy, we conduct an ablation study comparing the sigmoid-based density decay strategy with a cubic-based density decay strategy. We evaluate the performance on the LLaMA-60M model using the C4 dataset and report the perplexity. We show the number of density decay steps and a comparison between the cubic density decay that GraNet [40] and GMP [41] introduced and our proposed sigmoid density decay strategy. The results in the Table 13 show that the sigmoid-based decay consistently achieves lower PPL than the cubic-based decay in all the density decay steps, and both strategies outperform the baseline fixed sparsity (CHTs). These results confirm the motivation for introducing the sigmoid density decay strategy. The table below shows the PPL of CHTss with the two density decay strategies over different density decay steps.

**Curvature of the Density Decay.** In CHTss, the parameter $k$ dictates the pruning schedule's shape: lower $k$ values yield a smoother decay, while higher values create a sharper curve. We varied $k$ in 2, 4, 6, 8, 10 on LLaMA60M at 70% and 95% sparsity, using perplexity as the evaluation metric. Results on Table 14 show that overly sharp decay curves (larger $k$) cause instability. In the article, we used $k = 6$ for all experiments.

Table 13: Comparison of **CHTss** variants (Sigmoid- vs. Cubic-based) under different density decay steps. Baseline (CHTs) achieves 35.59. Lower values indicate better performance.

| Density Decay Steps | 1000 | 2000 | 3000 | 4000 | 5000 | 6000 | 7000 | 8000 | 9000 |
|---|---|---|---|---|---|---|---|---|---|
| **CHTss (Sigmoid-based)** | 35.20 | 35.19 | 35.24 | 34.63 | 34.68 | 34.63 | 34.46 | 34.50 | 34.39 |
| **CHTss (Cubic-based)** | 35.23 | 35.21 | 35.29 | 35.03 | 34.96 | 39.23 | 34.96 | 34.79 | 34.94 |

Table 14: Perplexity comparison of LLaMA60M under different initial sparsity and $k$ values at 70% and 95% sparsity levels. Lower is better.

| LLaMA60M (70% sparsity) | $k$=2.0 | $k$=4.0 | $k$=6.0 | $k$=8.0 | $k$=10.0 |
|---|---|---|---|---|---|
| 0.1 | **27.61** | 27.74 | 27.74 | 27.82 | 29.79 |
| 0.2 | **27.60** | 28.81 | 27.81 | 29.82 | 34.05 |
| 0.3 | 27.66 | 27.67 | 27.78 | 27.59 | **27.58** |
| 0.4 | 27.93 | 27.82 | 27.78 | 27.73 | **27.71** |
| 0.5 | 28.05 | 27.82 | **27.62** | 27.83 | 28.13 |

| LLaMA60M (95% sparsity) | $k$=2.0 | $k$=4.0 | $k$=6.0 | $k$=8.0 | $k$=10.0 |
|---|---|---|---|---|---|
| 0.1 | 37.00 | 36.02 | **35.86** | 36.42 | 36.59 |
| 0.2 | 36.83 | 36.08 | **35.42** | 38.39 | 35.87 |
| 0.3 | 36.67 | **35.30** | 35.53 | 35.67 | 35.78 |
| 0.4 | 35.71 | 35.48 | **35.42** | 35.44 | 35.63 |
| 0.5 | 36.76 | 35.70 | **35.59** | 35.72 | 36.05 |

**Removal Fraction $\zeta$.** This parameter determines the proportion of existing connections that are pruned during each pruning-regrowth cycle. It directly controls the amount of structural change introduced to the network at each step. In our sensitivity analysis on Table 15, we evaluated a range of $\zeta$ values (0.1, 0.2, 0.3, 0.4, 0.5) to assess their effect on the convergence and final performance of the CHTs model. The tests were performed on MLPs (CIFAR-10) at 90%, 95%, and 99% sparsities. Results are averaged over three seeds. When the sparsity is lower, we need a lower $\zeta$ to keep the model trained more stably, and when the sparsity is higher, a higher $\zeta$ is required to encourage the topological exploration. In all experiments involving MLP and Transformer, we use a $\zeta$ of 0.3, and for LLaMA models, we use 0.1 for all the experiments.

**CHTs vs. CHTss.** In most experiments, CHTss (with the sigmod density decay schedule) consistently outperforms CHTs. There are, however, a few counter-examples (e.g., LLaMA-1B at 70% target sparsity on OpenWebText) where CHTs slightly surpasses CHTss. To understand this outlier behavior, we first note that in the LLaMA-1B OpenWebText run the schedule only decreased density from 50% to 30%—a small change relative to the fixed 50% baseline—yielding very similar perplexities (14.66 vs. 15.15). When we extended the schedule to a much lower final density 5%, the advantage of CHTss became clear: CHTss reached 16.51 PPL versus 18.93 for CHTs. This aligns with our ITOP analysis: at 30% density, over 90% of candidate links appear during the evolution of both methods, but at 5% density CHTs touches only ∼20% of links while CHTss still explores ∼90%. In other words, density–decay confers a broader search over plausible connections precisely where search matters most—at high sparsity. We further corroborated this with a cross-scale study on six LLaMA sizes (20M(→)1B) at 70%, 90% , and 95% sparsity on Table 16: averaged validation perplexity differences are not significant at 70% (Wilcoxon (p=0.688)), but become significant in favor of CHTss at 90% and 95% (p=0.031 for both). Taken together, these results indicate that occasional wins of CHTs arise in regimes where the decay is shallow and the effective search space is already well covered; they do not suggest that the sigmoid density–decay strategy fails on larger models or particular tasks. Rather, its benefit scales with the target sparsity: the more aggressive the final sparsity, the more CHTss helps by exposing and testing a larger set of candidate links during training.

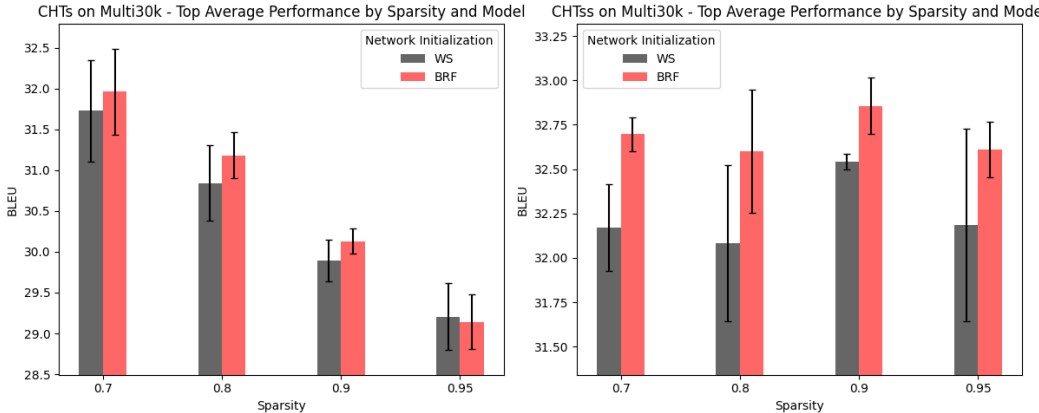

Figure 9: Top average BLEU for WS (grey) and BRF initialization methods on the Multi30k translation dataset of CHTs (left) and CHTss (right, with sigmoid decay) at different sparsity levels. Error bars denote the standard error across three seeds.

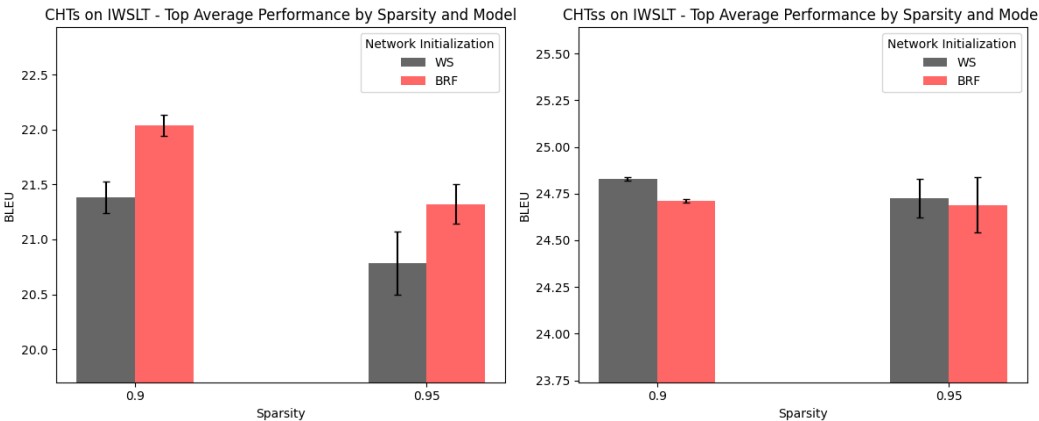

Figure 10: Top average BLEU for WS (grey) and BRF initialization methods on the IWSLT translation dataset of CHTs (left) and CHTss (right, with sigmoid decay) at different sparsity levels. Error bars denote the standard error across two seeds.

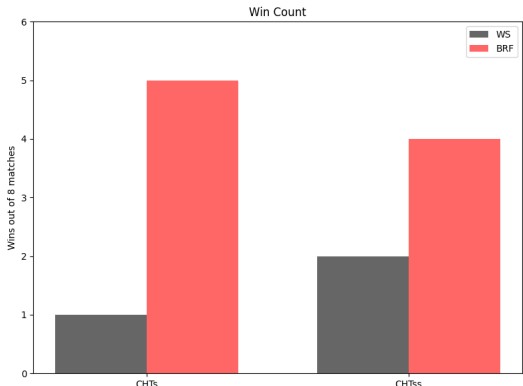

Figure 11: Win rates of BRF against WS over CHTs and CHTss models on different datasets (Multi30k and IWSLT) and different sparsities (0.9 and 0.95 for IWSLT and 0.7, 0.8, 0.9, 0.95 for Multi30k).

Table 15: Test accuracy (%) on CIFAR-10 under varying sparsity ratios and $\zeta$ values. Mean $\pm$ standard deviation over multiple runs. Bold indicates best per column.

| $\zeta$ | CIFAR-10 (90%) | CIFAR-10 (95%) | CIFAR-10 (99%) |
|---|---|---|---|
| 0.1 | **68.73 $\pm$ 0.27** | 69.28 $\pm$ 0.08 | 63.45 $\pm$ 0.04 |
| 0.2 | 68.63 $\pm$ 0.10 | **69.78 $\pm$ 0.09** | 67.30 $\pm$ 0.21 |
| 0.3 | 68.47 $\pm$ 0.10 | 69.37 $\pm$ 0.10 | **68.08 $\pm$ 0.02** |
| 0.4 | 67.90 $\pm$ 0.10 | 69.01 $\pm$ 0.08 | 67.83 $\pm$ 0.19 |
| 0.5 | 67.47 $\pm$ 0.14 | 68.56 $\pm$ 0.15 | 67.24 $\pm$ 0.14 |

Table 16: CHTs vs. CHTss across sparsity levels (validation perplexity; lower is better).

| Sparsity | Mean CHTs $\downarrow$ | Mean CHTss $\downarrow$ | $p$-value | Significance |
|---|---|---|---|---|
| 0.70 | 23.81 | 23.88 | 0.688 | Not significant |
| 0.90 | 28.47 | 26.51 | 0.031 | Significant |
| 0.95 | 31.89 | 29.19 | 0.031 | Significant |

# I Integral of Sigmoid Density Decay and Cubic Density Decay

In this section, we show the formula for the proposed Sigmoid Density Decay and the Cubic decay implemented in GraNet [40] and GMP [54].

For the Cubic function, it is formulated as:

$$s_t = s_f + (s_i - s_f)\left(1 - \frac{t - t_0}{n\Delta t}\right)^3, \tag{9}$$

where $t \in \{t_0, t_0 + \Delta t, \dots, t_0 + n\Delta t\}$, $s_i$ is the initial sparsity, $s_f$ is the target sparsity, $t_0$ is the starting epoch of gradual pruning, $t_f$ is the end epoch of gradual pruning, and $\Delta t$ is the pruning frequency.

Since our work focuses on MLP, Transformer, and LLMs, where FLOPs are linearly related to the density of the linear layers, the FLOPs of the whole training process are linearly related to the integral of the density function across the training time. The integral of the cubic decay function from $t_0$ to $t_f$

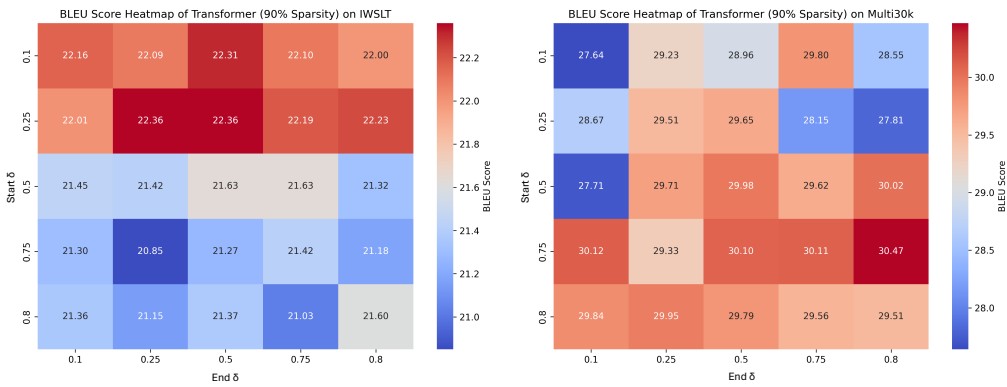

Figure 12: **Investigating the level of randomness in link removal strategies.** Top BLEU scores of the transformer model using CHTs with weight magnitude soft removal strategy, as the initial and final values of $\delta$ take values in $\{0.1, 0.25, 0.5, 0.75, 0.8\}$.

is:

$$\int_{t_0}^{t_f} (s_i - s_f) \left(1 - \frac{t - t_0}{n\Delta t}\right)^3 dt$$
$$= \frac{1}{4}(s_i - s_f)(t_f - t_0).$$

(10)

For the sigmoid decrease, the integral is:

$$\int_{t_0}^{t'_f} (s_i - s_f) \left(\frac{1}{1 + e^{-k\left(t - \frac{t'_f + t_0}{2}\right)}}\right) dt$$
$$= \frac{(s_i - s_f)(t'_f - t_0)}{2}.$$

(11)

To maintain consistency in the computational cost (FLOPs) during training compared to the cubic decay strategy, we reduce the number of steps in the sigmoid-based gradual density decrease by half.

## J    Historical weights

Inspired by GMP [54, 41], we incorporate historical weights into our CHTs and CHTss implementation. During training, we maintain a historical weight matrix that records previously learned weights throughout the training process. When CHTs and CHTss predict new links, we initialize them using their corresponding historical weights - specifically, the values they held before being pruned. In this way, CHTs and CHTss enable weight recovery with preserved memory, allowing the model to retain valuable prior information.

## K    Pseudo code of node-based CH link predictor

We present the pseudocode for the node-based CH2-L3 regrowth method in Algorithm 1, which comprises three main steps.

**Step 1:** We compute the sets of path length-2 (L2) neighbors from nodes $U$ to $U$ and from $V$ to $V$. The computational complexity of this step is

$$\mathcal{O}(m \langle d_m d_n \rangle + n \langle d_m d_n \rangle),$$

where $\langle d_m d_n \rangle$ denotes the average product of degrees $d_m$ and $d_n$ across all relevant node pairs. In the case of an ultra-sparse network (e.g., with average degree close to 1), this simplifies to $\mathcal{O}(m + n)$.

**Step 2:** We treat each destination node along the $UU$ and $VV$ paths as a potential common neighbor and compute the CH2-L3 score according to Equation (3) in the main text. This step has a time complexity of

$$\mathcal{O}(m^2 + n^2).$$

**Step 3:** We aggregate the CH2-L3 scores across all hop-3 nodes. This step requires

$$\mathcal{O}(mn \langle d_m \rangle + nm \langle d_n \rangle)$$

time. Under the ultra-sparse assumption, this further reduces to $\mathcal{O}(mn + nm)$.

Combining all steps, the overall time complexity of the node-based CH2-L3 regrowth procedure is

$$\mathcal{O}(mn \langle d_m \rangle + nm \langle d_n \rangle).$$

Since all operations can be implemented in a matrix-wise manner leveraging GPU acceleration assuming all matrices as dense, the total time complexity becomes

$$\mathcal{O}(nm^2 + mn^2).$$

**Algorithm 1** CH2–L3n

---

**Require:** Binary bipartite adjacency matrix $A_{UV} \in \{0, 1\}^{m \times n}$          $\triangleright$ $U$–to–$V$ edges
**Ensure:** Score matrix $S \in \mathbb{R}^{m \times n}$ with $S[i, j] > 0$ only if $A_{UV}[i, j] = 0$

1:  **function** CH2-L3N($A_{UV}$)
2:      $A_{VU} \leftarrow A_{UV}^{\top}$                                                    $\triangleright$ $V \to U$ edges
3:      $d_U \leftarrow$ row-sum($A_{UV}$)
4:      $d_V \leftarrow$ row-sum($A_{VU}$)
                                            $\triangleright$ **Step 1:** two–step paths inside each partition
5:      $UU \leftarrow A_{UV} A_{VU}$                                      $\triangleright$ $U \to V \to U$
6:      $VV \leftarrow A_{VU} A_{UV}$                                      $\triangleright$ $V \to U \to V$
                                               $\triangleright$ **Step 2:** CH2-L3n preparatory scores
7:      *init* $e_{UU} \leftarrow 0_{m \times m}$, $e_{VV} \leftarrow 0_{n \times n}$
8:      **for** $i \leftarrow 1$ **to** $m$ **do**
9:          **for** $j \leftarrow 1$ **to** $m$ **s.t.** $j \neq i$ **and** $UU[i, j] > 0$ **do**
10:             $ext \leftarrow d_U[j] - UU[i, j] - 1$
11:             $e_{UU}[i, j] \leftarrow \dfrac{UU[i, j] + 1}{ext + 1}$             $\triangleright$ # According to Equation (3)
12:          **end for**
13:      **end for**
14:      **for** $a \leftarrow 1$ **to** $n$ **do**
15:          **for** $b \leftarrow 1$ **to** $n$ **s.t.** $b \neq a$ **and** $VV[a, b] > 0$ **do**
16:             $ext \leftarrow d_V[b] - VV[a, b] - 1$
17:             $e_{VV}[a, b] \leftarrow \dfrac{VV[a, b] + 1}{ext + 1}$             $\triangleright$ # According to Equation (3)
18:          **end for**
19:      **end for**
                                               $\triangleright$ **Step 3:** final CH2–L3n scores
20:      *init* $S \leftarrow 0_{m \times n}$
21:      **for** $i \leftarrow 1$ **to** $m$ **do**
22:          **for** $a \leftarrow 1$ **to** $n$ **s.t.** $A_{UV}[i, a] = 0$ **do**
23:             $S_{UV} \leftarrow \sum\limits_{j=1}^{m} e_{UU}[i, j]\, A_{UV}[j, a]$
24:             $S_{VU} \leftarrow \sum\limits_{b=1}^{n} e_{VV}[a, b]\, A_{VU}[b, i]$
25:             $S[i, a] \leftarrow S_{UV} + S_{VU}$
26:          **end for**
27:      **end for**
28:      **return** $S$
29:  **end function**

---

# L   Extra results of LLaMA1b

**Language modeling.** We present a comparison of CHTs, CHTss, and fully connected network on language modeling tasks using the LLaMA-1B model on Table 17. The results clearly demonstrate that CHTs outperform the fully connected (FC) baseline at 70%, even at a high sparsity of 95%, CHTss achieves a perplexity of 16.51, which is remarkably close to the FC baseline.

**Zero-shot performance.** We compare CHTs, CHTss, and a fully connected (FC) network on zero-shot benchmarks, as summarized in Table 20. At 90% and 95% sparsity, CHTs often surpasses the FC baseline, suggesting improved generalization to unseen data. At 70% sparsity, CHTs is comparable to FC, whereas at higher sparsity (90–95%) it can exceed FC in accuracy. This pattern appears at odds with perplexity trends, so we provide additional analysis for readers evaluating pretrained models on GLUE/SuperGLUE in a zero-shot setting.

A key confound is label imbalance in several tasks: some datasets contain markedly skewed positive/negative distributions. In practice, we observed that models sometimes emit nearly the same

Table 17: **Validation perplexity of different dynamic sparse training (DST) methods on Open-WebText using LLaMA-1B across varying sparsity levels.**. Lower perplexity corresponds to better model performance. The performances that surpass the fully connected model are marked with "*".

| Sparsity | 0.7 | 0.9 | 0.95 |
|---|---|---|---|
| FC | | 14.62 | |
| CHTs | 14.53* | 17.14 | 18.93 |
| CHTss | 15.15 | 15.62 | 16.51 |

Table 18: Label imbalance in selected GLUE/SuperGLUE tasks (positive vs. negative counts and proportions).

| Dataset | Pos (count, %) | Neg (count, %) |
|---|---|---|
| CoLA | 721 (69.13%) | 322 (30.87%) |
| MRPC | 279 (68.38%) | 129 (31.62%) |
| QQP | 14,885 (36.82%) | 25,545 (63.18%) |
| BoolQ | 2,033 (62.17%) | 1,237 (37.83%) |

label for most inputs, yielding deceptively strong *accuracy* if that label matches the dataset majority class. Table 18 reports the class ratios for representative tasks.

To mitigate this issue, we report Matthews correlation coefficient (MCC) instead of accuracy. MCC is robust to class imbalance and better reflects the quality of binary predictions. Table 19 shows averaged MCC over GLUE/SuperGLUE under multiple sparsity regimes.

The 70% sparse models (MCC 0.028 for CHTs; 0.033 for CHTss) are on par with the dense FC baseline (MCC 0.031), but MCC declines as sparsity increases; at 95% sparsity, negative MCC indicates predictions are anticorrelated with ground truth (worse than random). We attribute part of this degradation to compute-limited pretraining budgets (e.g., up to 8.9B tokens for LLaMA-1B in our runs), which are insufficient to reach the regime where zero-shot performance is reliable at extreme sparsity. *Recommendation:* for zero-shot evaluation on imbalanced datasets, prefer MCC (or similarly balanced metrics) over raw accuracy, especially when comparing across sparsity levels and training budgets.

## M   Extra Related Works

More recently, BiDST has elegantly reframed DST as a bi-level optimization problem to co-optimize the weights and the sparsity mask simultaneously [57]. Other methods improve the training dynamics itself, such as AC/DC, which alternates between sparse and dense phases [58], or Powerpropagation, which introduces a sparsity-inducing weight reparameterization [59]. To enhance generalization in the often-chaotic loss landscape of sparse models, S2-SAM provides an efficient, plug-and-play sharpness-aware optimizer [60]. Another thrust of research adapts DST for specific architectural or hardware advantages. For example, SLaK utilizes dynamic sparsity to enable massive, high-performance kernels in CNNs [61]. CHASE provides a practical method to translate unstructured dynamic sparsity into hardware-friendly, channel-level sparsity [62]. Addressing network health, NEURREV proposes a mechanism to revitalize "dormant neurons" that can emerge during training [63].

## N   Experiments compute resources

All experiments were conducted on NVIDIA A100 80GB GPUs. MLP and Transformer models were trained using a single GPU, while LLaMA models were trained using eight GPUs in parallel.

Table 19: Average MCC on zero-shot GLUE/SuperGLUE. FC denotes the dense fully connected model.

| Metric | FC | 70% CHTs | 70% CHTss | 90% CHTs | 90% CHTss | 95% CHTs | 95% CHTss |
|---|---|---|---|---|---|---|---|
| AVG MCC | 0.031 | 0.028 | 0.033 | 0.031 | 0.022 | $-0.003$ | $-0.007$ |

Table 20: Zero-shot evaluation of LLaMA-1B across GLUE and SuperGLUE. ACC scores are shown. Values are mean ± sd over 5 seeds (lm-eval [56]). Red values indicate the top scores over models and sparsities for a benchmark.

| Dataset | FC | 70 % sparsity | | 90 % sparsity | | 95 % sparsity | |
|---|---|---|---|---|---|---|---|
| | | CHTs | CHTss | CHTs | CHTss | CHTs | CHTss |
| CoLA | 40.27 ± 1.52 | 38.83 ± 1.51 | 44.20 ± 1.54 | 50.53 ± 1.55 | 67.88 ± 1.45 | 68.84 ± 1.43 | 31.83 ± 1.44 |
| MNLI | 32.89 ± 0.47 | 32.77 ± 0.47 | 32.65 ± 0.47 | 32.83 ± 0.47 | 32.73 ± 0.47 | 32.91 ± 0.47 | 32.68 ± 0.47 |
| MRPC | 39.95 ± 2.43 | 40.44 ± 2.43 | 36.03 ± 2.38 | 66.42 ± 2.34 | 38.48 ± 2.41 | 67.65 ± 2.32 | 46.81 ± 2.47 |
| QNLI | 49.95 ± 0.68 | 50.96 ± 0.68 | 49.70 ± 0.68 | 50.36 ± 0.68 | 49.79 ± 0.68 | 49.26 ± 0.68 | 51.42 ± 0.68 |
| QQP | 47.94 ± 0.25 | 43.40 ± 0.25 | 49.12 ± 0.25 | 50.57 ± 0.25 | 48.22 ± 0.25 | 36.92 ± 0.24 | 41.70 ± 0.25 |
| RTE | 47.29 ± 3.01 | 46.93 ± 3.00 | 55.96 ± 2.99 | 46.57 ± 3.00 | 52.71 ± 3.01 | 52.35 ± 3.01 | 49.46 ± 3.01 |
| SST-2 | 65.71 ± 1.61 | 52.98 ± 1.69 | 49.66 ± 1.69 | 49.08 ± 1.69 | 53.21 ± 1.69 | 54.59 ± 1.69 | 49.08 ± 1.69 |
| WNLI | 50.70 ± 5.98 | 49.30 ± 5.98 | 47.89 ± 5.97 | 56.34 ± 5.93 | 50.70 ± 5.98 | 50.70 ± 5.98 | 40.85 ± 5.88 |
| Hellaswag | 28.98 ± 0.45 | 28.75 ± 0.45 | 28.70 ± 0.45 | 27.57 ± 0.45 | 28.13 ± 0.45 | 27.57 ± 0.45 | 27.53 ± 0.45 |
| Boolq | 41.07 ± 0.86 | 50.89 ± 0.87 | 48.62 ± 0.87 | 44.59 ± 0.87 | 47.46 ± 0.87 | 55.38 ± 0.87 | 52.14 ± 0.87 |
| CB | 48.21 ± 6.74 | 50.00 ± 6.74 | 50.00 ± 6.74 | 50.00 ± 6.74 | 50.00 ± 6.74 | 37.50 ± 6.53 | 48.21 ± 6.74 |
| Copa | 65.00 ± 4.79 | 62.00 ± 4.88 | 64.00 ± 4.82 | 66.00 ± 4.76 | 63.00 ± 4.85 | 60.00 ± 4.92 | 60.00 ± 4.92 |
| **Average** | 46.50 | 45.60 | 46.38 | 49.24 | 48.53 | 49.47 | 44.31 |

