# OpenReview forum: "Brain network science modelling of sparse neural networks enables Transformers and LLMs to perform as fully connected"
_NeurIPS.cc/2025/Conference — NeurIPS 2025 poster_

### Official Review · Reviewer_QgTr · 2025-07-01

**Clarity:** 2
**Significance:** 4
**Originality:** 3
**Rating:** 5
**Confidence:** 1

**Summary:**

This work improves brain-inspired dynamic sparse training (DST) by addressing key limitations of the Cannistraci-Hebb (CH) method. The experiments are comprehensive and the results are promising.

**Questions:**

NA.

**Ethical Concerns:**

["NO or VERY MINOR ethics concerns only"]

**Final Justification:**

Just keep my original judgement.

**Limitations:**

Yes.

**Paper Formatting Concerns:**

NA.

**Quality:**

3

**Strengths And Weaknesses:**

**Please note that I am not an expert in this field.**

Strengths
---

1. Compelling Bio-Inspired Foundation.
The motivation that translates brain network principles into dynamic sparse training (DST) is well-grounded. This bio-plausible method presents a credible way to advancing current AI.

2. Rigorous Technical Contributions.
The work demonstrates good mathematical and algorithmic innovation, including BRF initialization, GPU-optimized CH approximation, CHTs soft rule, CHTss framework and so on.

3. Extensive and Convincing Validation.
Experiments cover diverse domains, like vision classification, Transformers, language modeling, across recognized datasets. All results seem to be very promising.

4. Field-Advancing Potential.
This paradigm that leverages neural-biological principles, e.g., receptive field, Hebb plasticity, might catalyze a new generation of efficient AI, going beyond gradient-based optimization.

Weaknesses
---

1. Accessibility for Cross-Disciplinary Audiences.
Despite being technically rigorous, the manuscript assumes substantial prior knowledge of network neuroscience (e.g., Cannistraci-Hebb rules) and sparse training literature. Non-expert readers may struggle with it.

---

> ### Author Rebuttal · Authors · 2025-07-30
>
> We sincerely thank the Reviewer for the thoughtful and encouraging feedback. We are truly grateful for your recognition of the bio-inspired foundation, technical contributions, and the broad validation of our work.
>
> Regarding the noted weakness in accessibility for cross-disciplinary audiences, we fully understand the concern and agree that making the manuscript approachable to non-experts is important. In the revised version, we have created a new section in the Appendix named "glossary of network science and Cannistraci-Hebb rules" to introduce some basic notions and concepts of network science and CH rules involved in this article.
>
> We believe these changes will significantly improve the accessibility of the manuscript for broader audiences. We thank the Reviewer again for recognizing our work.

---

### Official Review · Reviewer_kVtz · 2025-07-03

**Clarity:** 2
**Significance:** 2
**Originality:** 3
**Rating:** 5
**Confidence:** 3

**Summary:**

The paper proposes a brain-inspired sparse training framework that initializes networks with receptive-field topologies, prunes and regrows connections using a gradient-free Cannistraci-Hebb rule, and gradually increases sparsity with a sigmoid schedule. This approach lets MLPs, Transformers, and a 130 M-parameter LLaMA variant retain or surpass dense accuracy while keeping only 1% to 30% of their original links.

**Questions:**

1. When training language models, up to what sparsity level can the proposed method still outperform the original dense model, or is it impossible to exceed the dense baseline at higher sparsities (over 50%)?
2. Does the proposed framework also work for convolutional architectures such as VGG and ResNet on standard image tasks?
3. For large model like LlaMA-1B, what is the cost of constructing and storing the BRF topology—both in terms of GPU memory and wall-clock time?
4. If the initial BRF sparsity is set too low or too high, does the sigmoid-decay curve need retuning? Could the two components be learned jointly?

**Ethical Concerns:**

["NO or VERY MINOR ethics concerns only"]

**Final Justification:**

The reviewer solved most of the questions. I will raise my score.

**Limitations:**

Please see weakness and question.

**Paper Formatting Concerns:**

None noted.

**Quality:**

3

**Strengths And Weaknesses:**

Strengths:
1. The framework is grounded in Cannistraci-Hebb network science, yielding a principled, gradient-free regrowth rule.
2. Probabilistic removal and regrowth mitigate overlap between deleted and added links, improving topology exploration and final accuracy.
3. The proposed framework demonstrates performance across MLPs, sequence-to-sequence Transformers and decoder-only LLMs.

Weakness:
1. Performance depends on several hyperparameters. For example, the soft-sampling temperature $\delta$, removal fraction $\zeta$, BRF randomness $r$, and sigmoid-decay curvature $k$—yet the paper provides little systematic analysis of how results vary with these settings.
2. The paper contains too many CHI-style notations, which makes it exhausting to read. Figure 1 is overly cluttered and hard to follow, and the subsections in Section 3 are not tightly connected.

3. Some important related works [1][2][3][4][5][6][7] need to be discussed.
[1] AC/DC: Alternating Compressed/DeCompressed Training of Deep Neural Networks, NeurIPS 2021
[2] Powerpropagation: A sparsity inducing weight reparameterisation, NeurIPS 2021
[3] Dynamic sparsity is channel-level sparsity learner, NeurIPS 2023
[4] More ConvNets in the 2020s: Scaling up Kernels Beyond 51x51 using Sparsity, ICLR2023
[5] NeurRev: Train Better Sparse Neural Network Practically via Neuron Revitalization, ICLR2024
[6] Advancing Dynamic Sparse Training by Exploring Optimization Opportunities, ICML2024
[7] A Single-Step, Sharpness-Aware Minimization is All You Need to Achieve Efficient and Accurate Sparse Training, NeurIPS 2024

---

> ### Author Rebuttal · Authors · 2025-07-30
>
> We sincerely thank the Reviewer for the thoughtful and encouraging feedback. Your positive comments on our work are greatly appreciated. Below, we respond point‑by‑point to the identified weaknesses and questions and describe the additional analyses now included in the revised manuscript.
>
> **Weakness 1: Little systematic analysis of how results vary with different parameter values**
>
> **Reply:** We fully understand the Reviewer's concern about the task-specific hyperparameter tuning that could probably boost the performance of our methods in each task. To clarify, our experiments are designed with minimal task-specific hyperparameter-tuning. The primary performance gains are achieved through the introduction of:
> - The novel **brain-inspired receptive field (BRF)** topological initialization (Figures 9–11).
> - The **soft sampling strategy** for link removal and regrowth, which balances exploration and exploitation (Figure 3, Table 12).
> - The **sigmoid-based density decay** that broadens the search space of connections, particularly at high sparsity (Tables 2, 3, 11, and 12).
> - The **node-based implementation** that provides a significant runtime speed-up (Figure 2, Table 12).
>
> These core innovations introduce several additional hyperparameters. However, we found that the model performance remains relatively stable across a reasonable range of values. To substantiate this, we added a new section in the Appendix presenting sensitivity analyses of these hyperparameters. Due to space limitations, we summarize the key findings here:
>
> - Soft sampling factor **δ**. As shown in **Figure 7**, we observe that higher initial temperatures perform better on simple benchmarks, whereas lower initial values are preferable on more complex tasks like IWSLT. Across our LLM experiments, we used a δ starting at 0.5 and ending at 0.9.
> - The BRF initialisation randomness parameter **r**. Results in **Tables 9 and 10** indicate that values of 𝑟 in [0, 0.3] yield quite stable performance.
> - Curvature of the Density Decay Curve **k**. In CHTss, the parameter k dictates the pruning schedule's shape: lower k values yield a smoother decay, while higher values create a sharper curve. We varied k in {2, 4, 6, 8, 10} on LLaMA60M at 70% and 95% sparsity, using perplexity as the evaluation metric. Results show that overly sharp decay curves (larger k) cause instability. In the article, we used k = 6 for all experiments.
>
> **LLaMA60M** (70% sparsity)
> |Initial Sparsity|k=2.0|k=4.0|k=6.0|k=8.0|k=10.0|
> |--:|--:|--:|--:|--:|--:|
> |0.1|**27.61**|27.74|27.74|27.82|29.79|
> |0.2|**27.60**|28.81|27.81|29.82|34.05|
> |0.3|27.66|27.67|27.78|27.59|**27.58**|
> |0.4|27.93|27.82|27.78|27.73|**27.71**|
> |0.5|28.05|27.82|**27.62**|27.83|28.13|
>
> **LLaMA60M** (95% sparsity)
> |Initial Sparsity|k=2.0|k=4.0|k=6.0|k=8.0|k=10.0|
> |--:|--:|--:|--:|--:|--:|
> |0.1|37.00|36.02|**35.86**|36.42|36.59|
> |0.2|36.83|36.08|**35.42**|38.39|35.87|
> |0.3|36.67|**35.30**|35.53|35.67|35.78|
> |0.4|35.71|35.48|**35.42**|35.44|35.63|
> |0.5|36.76|35.70|**35.59**|35.72|36.05|
>
> - Removal Fraction (**ζ**) — This parameter determines the proportion of existing connections that are pruned during each pruning-regrowth cycle. It directly controls the amount of structural change introduced to the network at each step. In our sensitivity analysis, we evaluated a range of ζ values (0.1, 0.2, 0.3, 0.4, 0.5) to assess their effect on the convergence and final performance of the CHTs model. The tests were performed on MLPs (CIFAR-10) at 90%, 95%, and 99% sparsities. Results are averaged over three seeds. When the sparsity is lower, we need a lower **ζ** to keep the model trained more stably, and when the sparsity is higher, a higher **ζ** is required to encourage the topological exploration. In all experiments involving MLP and Transformer, we use a **ζ** of 0.3, and for LLaMA models, we use 0.1 for all the experiments.
>
> |ζ| CIFAR-10 (90%)| CIFAR-10 (95%)| CIFAR-10 (99%)|
> |-:|:-|:-|:-|
> |0.1 | **68.73 ± 0.27** | 69.28 ± 0.08 | 63.45 ± 0.04 |
> |0.2 | 68.63 ± 0.10 | **69.78 ± 0.09** | 67.30 ± 0.21 |
> |0.3 | 68.47 ± 0.10 | 69.37 ± 0.10 | **68.08 ± 0.02** |
> |0.4 | 67.90 ± 0.10 | 69.01 ± 0.08 | 67.83 ± 0.19 |
> |0.5 | 67.47 ± 0.14 | 68.56 ± 0.15 | 67.24 ± 0.14 |
>
> **Weakness 2: too many CHI-style notations. Figure 1 is overly cluttered. The subsections in Section 3 are not tightly connected.**
>
> **Reply:** While the term “CHI-style notations” is not entirely clear to us, we interpret the concern as referring to an excessive use of mathematical symbols, abbreviations, or formalism that may hinder readability. In response, we have reduced the number of mathematical notations and abbreviations where possible. For example, we revised the paragraph on ELM in Subsection 3.1 (L141-151) as follows:
> ``Epitopological Local Minima occur when a large fraction of the newly added connections are the same ones that were just pruned in the preceding step. Such a cycle of removal and regrowth indicates that the model is trapped, preventing it from effectively exploring more optimal network topologies.
> This definition is crucial for understanding the challenges of the original Cannistraci-Hebb Training (CHT), where the rate of overlap between removed and regrown links becomes significantly high within just a few epochs, leading to a rapid convergence toward an ELM. Previously, CHT implemented a topological early stop strategy to avoid predicting the same links iteratively. However, it will stop the topological exploration very fast and potentially trap the model within the ELM."
>
> Additionally, we reordered subsections 3.1 and 3.2 to align with the sequential steps of our DST method and increased the font size of the content in Figure 1.
>
> **Weakness 3: Missing related works**
>
> **Reply:** We thank the Reviewer for pointing out these omissions. We have now integrated the suggested references into the Related Works section. Specifically, we added the following paragraph at **L113**:
> ``More recently, BiDST has elegantly reframed DST as a bi-level optimization problem to co-optimize the weights and the sparsity mask simultaneously. Other methods improve the training dynamics itself, such as AC/DC, which alternates between sparse and dense phases, or Powerpropagation, which introduces a sparsity-inducing weight reparameterization. To enhance generalization in the often-chaotic loss landscape of sparse models, S2-SAM provides an efficient, plug-and-play sharpness-aware optimizer. Another thrust of research adapts DST for specific architectural or hardware advantages. For example, sLaK utilizes dynamic sparsity to enable massive, high-performance kernels in CNNs. Chase provides a practical method to translate unstructured dynamic sparsity into hardware-friendly, channel-level sparsity. Addressing network health, NeurRev proposes a mechanism to revitalize "dormant neurons" that can emerge during training."
>
> We will also ensure these references are included in the final camera-ready version.
>
> **Question 1: At what sparsity level can the proposed method still outperform the dense baseline?**
>
> **Reply:** To address the reviewer's question, we have run additional tests of the CHTs model across two sizes of models, LLaMA models (60M and 1B), with sparsities from 50% to 95%. The table below compares the evaluated models (PPL as evaluation).
>
> **LLaMA60M** (Dense baseline: 26.56)
> |Model|50%|60%|70%|80%|90%|95%|
> |-:|:-|:-|:-|:-|:-|:-|
> |CHTs|27.24|27.60|28.12|29.84|33.03|36.47|
> |CHTss|27.24|27.65|27.62|29.00|31.42|35.10|
>
> **LLaMA1B** (Dense baseline: 14.62)
> |Model|60%|70%|90%|95%|
> |-:|:-|:-|:-|:-|
> |CHTs|14.48* |14.53*|17.14|18.93|
> |CHTss|14.77|15.15|15.62|16.51|
>
> The performance clearly shows that it is tough for a sparse method to outperform the dense model when the model is small, but on a larger model like LLaMA1B, CHTs outperforms the dense model with sparsity lower than 70%.
>
> **Question 2: Application to Convolutional architectures**
>
> **Reply:** As noted in the Limitation section, we are actively investigating the application of topology-based dynamic sparse training to CNNs. Now we have some preliminary results on MNIST using LeNet. The results of CHTs are close to those of the dense model and show improvements over the non-evolving static method. However, since implementing CHTs on CNNs involves several additional innovations, we may prefer to present these findings in a separate article.
> | |Accuracy|
> |-:|-:|
> |Dense|98.89|
> |50% CHTs|98.75|
> |50% Static sparse|98.45|
>
> To show that our methods are also able to handle vision tasks, we evaluated ViT-Base on ImageNet, where our method achieved higher accuracy than the dense baseline at up to 80% sparsity:
>
> **Dense ViT-Base: Accuracy 78.5**
> |Sparsity|Accuracy|
> |-:|-:|
> |0.6|79.88*|
> |0.7|79.38*|
> |0.8|79.05*|
> |0.9|77.54|
> |0.95|71.61|
>
> We will include these results in the final version.
>
> **Question 3: the cost of constructing and storing the BRF topology**
>
> **Reply:** The BRF is generated layer-by-layer and directly merged into the weight matrix to form the sparse weights. Therefore, the memory usage is negligible (<1GB). The generation time per layer (size 2048×2048) is approximately 3 seconds, and for LLaMA1B, the total cost is about 590s (6m).
>
> **Question 4: If the initial BRF sparsity is set too low or too high, does the sigmoid-decay curve need retuning? Could the two components be learned jointly?**
>
> **Reply:** This is a very sound question. We agree that the interplay between the initial sparsity and the sigmoid decay schedule can influence the training dynamics, especially in the early stages. To explore this, please refer to the Table and analysis in **Reply to Weakness 1** that compares different initial sparsities and corresponding values of the curvature parameter **k** used in the sigmoid decay.
>
> Currently, we do not jointly learn the model’s sparsity and the decay schedule. However, we truly value this insight and will consider joint optimization or learnable scheduling as a direction for future work.

---

> ### Comment · Reviewer_kVtz · 2025-08-06
>
> Thank you for the detailed rebuttal. I will raise my positive score from 4 to 5.

---

### Official Review · Reviewer_QCkw · 2025-07-03

**Clarity:** 1
**Significance:** 3
**Originality:** 2
**Rating:** 4
**Confidence:** 3

**Summary:**

This paper proposed a new Dynamic Sparsity Training (DST) method called Cannistraci-Hebb training soft rule (CHTs) to overcome the training complexity problem of existing methods, and achieved equivalent even superior results compared to fully connected networks.

**Questions:**

Questions:
1. Can CHTss be implemented efficiently on GPU to reduce training time compared to the backpropagation algorithm?
2. To my understanding, the basic building block of artificial neural networks is fully connected layers, which should only contain the input and output nodes. So why does the author put a network consists of 2 fully connected layers in Fig.1, while it is not widely used in Transformers?
3. The result of Llama-130M on GLUE dataset seems to underperform older models such as BERT[1]. Why does it happen and does it have an impact of the correctness of the conclusion?

[1] Devlin, Jacob et al. “BERT: Pre-training of Deep Bidirectional Transformers for Language Understanding.” North American Chapter of the Association for Computational Linguistics (2019).

**Ethical Concerns:**

["NO or VERY MINOR ethics concerns only"]

**Final Justification:**

Thank you to the authors for their response, which has adequately addressed my concerns. I will adjust my score accordingly. Additionally, I recommend that the authors include their reply to Weakness 3 in the appendix for the benefit of future readers.

**Limitations:**

yes

**Paper Formatting Concerns:**

No.

**Quality:**

3

**Strengths And Weaknesses:**

Strengths:
1. This paper proposes a modification to a DST method and significantly increases the accuracy and training efficiency of the method on various tasks.
2. The result of the paper suggests that sparse networks may achieve equivalent performance to dense networks.

Weakness:
1. The abstract of the article is a bit too long and somewhat difficult to understand. For example, the author mentioned bipartite receptive field (BRF) twice, but I’m still confused about the benefit of this module, and in line 26, the author said that “we propose a sigmoid-based gradual density decay strategy”, but it still seems unclear to me what is the motivation for designing such a strategy. I hope that the author could make the abstract shorter and clearer in logic to help more readers understand the core of this paper.
2. In most of the experiment, CHTss outperforms CHTs, but in the experiment of Llama-130M on SuperGLUE (Table 4) and Llama-1b 70% sparsity (Table 3) on OpenWebtext, CHTs seems to outperform CHTss. Could this mean that sigmoid density decrease is likely to fail on larger networks or some specific tasks? I hope that the author could provide some insights or theoretical analysis on this phenomenon.
3. The experiment in the supplementary material shows that CHTs performs close to fully connected network at sparsity level 70%, and surpass the fully connected networks when the sparsity level increases to 90% and 95%. This seems counter-intuitive to me as performance generally decreases when sparsity increases. Could the author provide a theoretical explanation for this phenomenon?

---

> ### Author Rebuttal · Authors · 2025-07-30
>
> We thank the Reviewer's insightful suggestions. We could see that the main concern raised by the reviewer is about the zero-shot performance on GLUE and SuperGLUE. We will give our clarification one by one.
>
> **Weakness 1: The unclear Abstract and the motivation for the introduction of the sigmoid density decay.**
>
> **Reply:** Thank you for your constructive suggestions. We have revised the abstract to address your concerns:
> - We removed the sentence in Line 23 that mentioned BRF twice to improve clarity and avoid redundancy.
> - We clarified the motivation behind the sigmoid-based gradual density decay strategy by adding the following sentence: “Additionally, we propose a sigmoid-based gradual density decay strategy to enable a smoother decay during the pruning stage, leading to an enhanced framework referred to as CHTss.”
> - To further demonstrate the benefit of this strategy, we conducted an ablation study comparing the sigmoid-based density decay strategy with a cubic-based density decay strategy. We evaluated the performance on the LLaMA-60M model using the C4 dataset and report the perplexity (PPL; lower is better). We show the number of density decay steps and a comparison between the cubic density decay that GraNet and GMP introduced and our proposed sigmoid density decay strategy. The results in the table below show that **the sigmoid-based decay consistently achieves lower PPL than the cubic-based decay in all the density decay steps**, and **both strategies outperform the baseline fixed sparsity (CHTs)**. These results confirm the motivation for introducing the sigmoid density decay strategy. The table below shows the PPL of CHTss with the two density decay strategies over different density decay steps.
>
> Baseline: CHTs 35.59
> |Density Decay Steps|1000|2000|3000|4000|5000|6000|7000|8000|9000|
> |-|-|-|-|-|-|-|-|-|-|
> |CHTss (Sigmoid-based)|35.20|35.19|35.24|34.63|34.68|34.63|34.46|34.50|34.39|
> |CHTss (Cubic-based)|35.23|35.21|35.29|35.03|34.96|39.23|34.96|34.79|34.94|
>
> **Weakness 2: Why in some cases CHTs outperform CHTss? Could this mean that sigmoid density decrease is likely to fail on larger networks or some specific tasks?**
>
> **Reply:** We thank the reviewer for this insightful question. It is true that in a few cases (e.g., LLaMA-130M on SuperGLUE and LLaMA-1B at 70% sparsity on OpenWebtext), CHTs slightly outperforms CHTss. In the reply of **Weakness 3**, we have discussed the zero-shot performance. Regarding the OpenWebText results:
> - In the LLaMA-1B experiment, the density decay schedule reduced density only from 50% to 30%, which is a relatively small change compared to the fixed 50% sparsity baseline. As a result, the performance difference between CHTs and CHTss in this setup is minimal (14.66 vs. 15.15 PPL). However, when we extended the experiment to a final density of 5%, CHTss achieved a PPL of 16.51, significantly outperforming CHTs (18.93). This suggests that the benefit of CHTss becomes more pronounced at higher sparsity levels by trying more possible links during training. By computing the ITOP$^1$ [1], we find that at 30% density, more than 90% of links appear in the evolution of the models for both CHTs and CHTss, while at 5% density, CHTs only involves around 20%, while CHTss still involves around 90%. This gives an explanation of why the density decay strategy is better when the final density is lower.
>
> To support our hypothesis, we ran additional experiments on six LLaMA model sizes (20 M → 1 B) for both CHTs and CHTss at 70 %, 90 %, and 95 % sparsity. For each sparsity level we computed the mean validation perplexity (lower is better) and used a two‑sided Wilcoxon signed‑rank test on the six paired values to assess whether the difference between CHTs and CHTss is statistically significant. As the table below shows, CHTss outperforms CHTs significantly at the higher sparsities (90 % and 95 %; p = 0.031 each), whereas at 70 % sparsity the difference is not significant (p = 0.688). We have added this analysis in the **Appendix** to clarify this.
> |Sparsity|Mean CHTs|Mean CHTss|p-value|Significance|
> |-|-|-|-|-|
> |0.7|23.81|23.88|0.688|Not significant|
> |0.9|28.47|26.51|0.031|Significant|
> |0.95|31.89|29.19|0.031|Significant|
>
> **Weakness 3: Why are the zero-shot evaluations of LLaMA1b, 90% sparsity, and 95% sparsity better than fully connected networks and 70% sparsity?**
>
> **Reply:** Thank you for this insightful comment. We agree that the result appears counter-intuitive, this phenomenon is probably caused by the imbalance of the datasets.
> -  In some of the datasets in GLUE and SuperGLUE, the positive and negative labels are very imbalanced. When we printed the output of the answer for each model, we found that they can usually only give the same reply for all the questions. Therefore, the accuracy is computed mainly based on the ratio of the different labels in the dataset.
> |Dataset|Pos (count, %)| Neg (count, %)|
> |-|-|-|
> |cola|721(69.13%)| 322 (30.87%)|
> |mrpc|279 (68.38%)| 129 (31.62%)|
> |qqp|14,885 (36.82%)| 25,545 (63.18%)|
> |boolq|2,033 (62.17%) |1,237 (37.83%)|
> - To address this, instead of using accuracy as the metrics, we switched to the Matthews correlation coefficient (MCC)$^2$, which is a measurement to address the imbalance of the dataset. We have the following results on GLUE and SuperGLUE on zero-shot evaluation:
> |Metric|FC|70% CHTs|70% CHTss|90% CHTs|90% CHTss|95% CHTs|95% CHTss|
> |-|-|-|-|-|-|-|-|
> |AVG of MCC|0.031|0.028|0.033|0.031|0.022|-0.003|-0.007|
>
> The results show that models with 70% sparsity achieve performance (MCC of 0.028 for CHTs and 0.033 for CHTss) comparable to the dense, fully connected (FC) model (MCC of 0.031). However, performance degrades significantly as sparsity increases. At 95% sparsity, the models yield negative MCC scores. A negative MCC indicates that the models' predictions are often the opposite of the correct answer, performing worse than random guessing.
>
> **Question 1: Can CHTss be implemented efficiently on GPU to reduce training time compared to the backpropagation algorithm?**
>
> **Reply:** Thank you for the question. We would like to clarify that the evolution processes of both CHTs and CHTss are already implemented efficiently on GPU, involving only a few matrix multiplications.
>
> If the reviewer’s question refers to a comparison of the evolution time with other methods such as GraNet (which uses gradients for regrowth), both of the evolution methods are quite fast. For example, on LLaMA-1B, the average evolution step time for GraNet is 1.92s, while for CHTs and CHTss it is 1.27s. The evolution time of both methods is eligible.
>
> **Question 2: Why does the author put a network consists of 2 fully connected layers in Fig.1, while it is not widely used in Transformers?**
>
> **Reply:** Thank you for the question. We provide three clarifications:
> - One of the key components of CHTs is the **network percolation** step, which is only meaningful when there are multiple layers. This step enables the removal of inactive nodes, allowing the algorithm to induce both unstructured and structured sparsities. Therefore, we intentionally used two layers in Figure 1 to clearly illustrate how this process operates.
> - Although the figure uses two layers for illustration, such structures are in fact widely used in neural networks, including Transformers. For instance, in GPT and Transformer, the Feed-Forward Network (FFN) sublayer consists of two consecutive linear layers. In LLaMA specifically, the gate_proj and up_proj layers operate in parallel and are followed by the down_proj layer.
> - As mentioned in **Appendix F**, to also introduce network percolation in Transformers, we also treat the q and k projections as two continuous layers, the v and o projections as another two continuous layers, and similarly, the FFN layers as consecutive layers for applying the network percolation step.
>
> To address the reviewer’s concern and avoid any ambiguity, we have added a sentence in the caption of **Figure 1** to clarify this:
> “As explained in Appendix F, the two-layer illustration also reflects how we treat q/k, v/o, and FFN layers as consecutive layers when applying the network percolation step in Transformers.”
>
> **Question 3: Llama-130M underperforms older models such as BERT[1] on GLUE. Why?**
>
> **Reply:** Thank you for raising this point. We believe the observed difference is primarily due to differences in training budgets and evaluation settings and an inherent advantage of BERT model:
> - In our article, the LLaMA-130M model was only pretrained for 40k steps, while in the original BERT paper, the model was pretrained for 1M steps, which is 25× more iterations. This naturally leads to a performance gap.
> - The GLUE results reported in the BERT paper are based on fine-tuned models, while in our work, we report zero-shot evaluation.
> - More importantly, BERT is the bidirectional encoder, so each token attends to the entire sequence—an advantage for classification and other tasks that require global context. In contrast, decoder‑only models such as LLaMA process tokens left‑to‑right, making them naturally stronger at generative tasks like open‑ended text generation.
>
> $^1$: ITOP computes how many links have been involved during the whole dynamic sparse training process.
>
> $^2$: MCC gives a single score between -1 and 1. Towards 1 indicates perfect prediction, towards -1 indicates total disagreement, and 0 means predictions are random guessing.
>
> [1] Do we actually need dense over-parameterization? in-time over-parameterization in sparse training. S Liu, L Yin, DC Mocanu, M Pechenizkiy. ICML 2021

---

> > ### Comment · Reviewer_QCkw · 2025-08-05
> >
> > Thank you to the authors for their response, which has adequately addressed my concerns. I will adjust my score accordingly. Additionally, I recommend that the authors include their reply to Weakness 3 in the appendix for the benefit of future readers.

---

### Official Review · Reviewer_tKGL · 2025-07-03

**Clarity:** 3
**Significance:** 3
**Originality:** 3
**Rating:** 4
**Confidence:** 4

**Summary:**

This study introduces brain-inspired methods to enhance dynamic sparse training (DST) in artificial neural networks (ANNs), addressing limitations in existing approaches like the Cannistraci-Hebb training (CHT). The proposed bipartite receptive field (BRF) initializes sparse connectivity, while a GPU-friendly CHT approximation reduces computational complexity. The CHT soft rule (CHTs) improves link sampling flexibility, and a sigmoid-based gradual density decay (CHTss) further optimizes sparsity. Experiments demonstrate that BRF outperforms prior models, CHTs achieves better accuracy than dense networks with just 1% connectivity in MLPs, and CHTss excels in Transformer-based translation tasks at 5% sparsity. At 30% connectivity, both methods surpass other DST techniques in language modeling and zero-shot tasks, proving efficient and scalable for ultra-sparse ANNs.

**Questions:**

See Weaknesses.
1. The paper acknowledges that hardware support for unstructured sparsity is not yet widespread, which limits the immediate practical deployment of the method. This could delay real-world adoption.
2. The method introduces several hyperparameters. While some sensitivity analysis is provided, it is not exhaustive. The optimal settings might vary across tasks, which could complicate deployment.
3. The paper claims full reproducibility, but some results are based on limited runs due to resource constraints. This could affect the reliability of the reported improvements.

**Ethical Concerns:**

["NO or VERY MINOR ethics concerns only"]

**Final Justification:**

I will maintain the positive score.

**Limitations:**

Yes

**Quality:**

3

**Strengths And Weaknesses:**

Strengths:
1. The paper introduces a novel brain-inspired method (CHTss) for dynamic sparse training (DST) that leverages the Cannistraci-Hebb theory, offering a unique perspective on sparse neural network training. The integration of topological principles from neuroscience is a significant contribution.
2. The proposed method achieves impressive results, outperforming fully connected networks in various tasks (e.g., MLPs at 99% sparsity, Transformers at 5% density). The empirical results are well-documented and supported by extensive experiments across multiple datasets and architectures.
3. The paper addresses the high computational complexity of previous methods
4. The study evaluates the method on diverse tasks (image classification, machine translation, language modeling) and architectures (MLPs, Transformers, LLaMA), demonstrating broad applicability.

Weaknesses:
1. The paper acknowledges that hardware support for unstructured sparsity is not yet widespread, which limits the immediate practical deployment of the method. This could delay real-world adoption.
2. The method introduces several hyperparameters. While some sensitivity analysis is provided, it is not exhaustive. The optimal settings might vary across tasks, which could complicate deployment.
3. The paper claims full reproducibility, but some results are based on limited runs due to resource constraints. This could affect the reliability of the reported improvements.

---

> ### Author Rebuttal · Authors · 2025-07-28
>
> We sincerely thank the Reviewer for the thoughtful and encouraging review of our work, particularly for recognizing the novelty of our brain-inspired method and the broad applicability across multiple architectures and tasks. We will reply to the weaknesses you raised in detail.
>
> **Weakness 1: Hardware support for unstructured sparsity is not yet widespread. This could delay real-world adoption.**
>
> **Reply:** We appreciate the reviewer’s comment and would like to respond from two perspectives:
> - Our method (CHTs) is ideal for edge AI and robotics, where GPU computing isn't feasible due to hardware and power limits. It generates lightweight, sparse models perfect for micro-drones, micro-robots, and other embodied edge devices, enabling efficient real-time training without GPUs.
> - While current GPUs favour dense computations, new hardware from companies like Cerebras is emerging to efficiently support unstructured sparsity (Lie, Sean. "Harnessing the Power of Sparsity for Large GPT AI Models." Cerebras Systems Blog (2022)). Our method is future-proof, aiming to fully leverage dynamic sparse training and inference as this hardware becomes more accessible.
>
> **Weakness 2: The method introduces several hyperparameters. While some sensitivity analysis is provided, it is not exhaustive. The optimal settings might vary across tasks, which could complicate deployment.**
>
> **Reply:** We fully understand the Reviewer's concern about the task-specific hyperparameter tuning that could probably boost the performance of our methods in each task. To clarify, our experiments are designed with minimal task-specific hyperparameter-tuning. The primary performance gains are achieved through the introduction of:
> - The novel **brain-inspired receptive field (BRF)** topological initialization (Figures 9–11).
> - The **soft sampling strategy** for link removal and regrowth, which balances exploration and exploitation (Figure 3, Table 12).
> - The **sigmoid-based density decay** that broadens the search space of connections, particularly at high sparsity (Tables 2, 3, 11, and 12).
> - The **node-based implementation** that provides a significant runtime speed-up (Figure 2, Table 12).
>
> These core innovations introduce several additional hyperparameters. However, we found that the model performance remains relatively stable across a reasonable range of values. To substantiate this, we added a new section in the Appendix presenting sensitivity analyses of these hyperparameters. Due to space limitations, we summarize the key findings here:
>
> - Soft sampling factor **δ**. As shown in **Figure 7**, we observe that higher initial temperatures perform better on simple benchmarks, whereas lower initial values are preferable on more complex tasks like IWSLT. Across our LLM experiments, we used a δ starting at 0.5 and ending at 0.9.
> - The BRF initialisation randomness parameter **r**. Results in **Tables 9 and 10** indicate that values of 𝑟 in [0, 0.3] yield quite stable performance.
> - Curvature of the Density Decay Curve **k**. In CHTss, the parameter k dictates the pruning schedule's shape: lower k values yield a smoother decay, while higher values create a sharper curve. We varied k in {2, 4, 6, 8, 10} on LLaMA60M at 70% and 95% sparsity, using perplexity as the evaluation metric. Results show that overly sharp decay curves (larger k) cause instability. In the article, we used k = 6 for all experiments.
>
> **LLaMA60M** (70% sparsity)
> |Initial Sparsity|k=2.0|k=4.0|k=6.0|k=8.0|k=10.0|
> |--:|--:|--:|--:|--:|--:|
> |0.1|**27.61**|27.74|27.74|27.82|29.79|
> |0.2|**27.60**|28.81|27.81|29.82|34.05|
> |0.3|27.66|27.67|27.78|27.59|**27.58**|
> |0.4|27.93|27.82|27.78|27.73|**27.71**|
> |0.5|28.05|27.82|**27.62**|27.83|28.13|
>
> **LLaMA60M** (95% sparsity)
> |Initial Sparsity|k=2.0|k=4.0|k=6.0|k=8.0|k=10.0|
> |--:|--:|--:|--:|--:|--:|
> |0.1|37.00|36.02|**35.86**|36.42|36.59|
> |0.2|36.83|36.08|**35.42**|38.39|35.87|
> |0.3|36.67|**35.30**|35.53|35.67|35.78|
> |0.4|35.71|35.48|**35.42**|35.44|35.63|
> |0.5|36.76|35.70|**35.59**|35.72|36.05|
>
> - Removal Fraction (**ζ**) — This parameter determines the proportion of existing connections that are pruned during each pruning-regrowth cycle. It directly controls the amount of structural change introduced to the network at each step. In our sensitivity analysis, we evaluated a range of ζ values (0.1, 0.2, 0.3, 0.4, 0.5) to assess their effect on the convergence and final performance of the CHTs model. The tests were performed on MLPs (CIFAR-10) at 90%, 95%, and 99% sparsities. Results are averaged over three seeds. When the sparsity is lower, we need a lower **ζ** to keep the model trained more stably, and when the sparsity is higher, a higher **ζ** is required to encourage the topological exploration. In all experiments involving MLP and Transformer, we use a **ζ** of 0.3, and for LLaMA models, we use 0.1 for all the experiments.
>
> |ζ| CIFAR-10 (90%)| CIFAR-10 (95%)| CIFAR-10 (99%)|
> |-:|:-|:-|:-|
> |0.1 | **68.73 ± 0.27** | 69.28 ± 0.08 | 63.45 ± 0.04 |
> |0.2 | 68.63 ± 0.10 | **69.78 ± 0.09** | 67.30 ± 0.21 |
> |0.3 | 68.47 ± 0.10 | 69.37 ± 0.10 | **68.08 ± 0.02** |
> |0.4 | 67.90 ± 0.10 | 69.01 ± 0.08 | 67.83 ± 0.19 |
> |0.5 | 67.47 ± 0.14 | 68.56 ± 0.15 | 67.24 ± 0.14 |
>
> **Weakness 3: The paper claims full reproducibility, but some results are based on limited runs due to resource constraints. This could affect the reliability of the reported improvements.**
>
> **Reply:** We have now added results averaged over three random seeds for all Transformer experiments in **Table 2**. These updated results provide a more reliable estimate of performance and confirm the robustness of our reported improvements. Tests on the CHT model were run on a unique seed due to time limitations.
> | Method | Multi30k (95%) | Multi30k (90%) | IWSLT (95%) | IWSLT (90%) | WMT (95%) | WMT (90%) |
> | :--- | :--- | :--- | :--- | :--- | :--- | :--- |
> | FC | 31.38 ± 0.38 | | 24.48 ± 0.30 | | 25.49± 0.15 | |
> | SET | 28.99 ± 0.28| 29.73 ± 0.10 | 18.53 ± 0.05 | 20.13 ± 0.08 | 20.19 ± 0.12| 21.61 ± 0.28|
> | RigL| 29.94 ± 0.27| 30.26 ± 0.34 | 20.53 ± 0.21 | 21.52 ± 0.15 | 20.71 ± 0.21| 22.22 ± 0.10|
> | CHT| 27.79| 28.38| 18.59| 19.91| 19.03| 21.08|
> | CHTs| 28.94 ± 0.57| 29.81 ± 0.37 | 21.15 ± 0.10 | 21.92 ± 0.17 | 20.94 ± 0.63 | 22.40 ± 0.06 |
> | MEST$_{EM\\&S}$ | 28.89 ± 0.26| 30.04 ± 0.52 | 19.56 ± 0.10 | 21.05 ± 0.21 | 20.70 ± 0.07| 22.22 ± 0.10|
> | GMP| 30.51 ± 0.82| 30.49 ± 0.40 | 22.76 ± 0.82 | 22.82 ± 0.53 | 22.47 ± 0.10| 23.37 ± 0.08|
> | GraNet| 31.31 ± 0.31| 31.62 ± 0.48* | 22.53 ± 0.12 | 22.43 ± 0.09 | 22.51 ±0.21| 23.46 ± 0.09|
> | CHTss| **32.03 ± 0.29*** | **32.86 ± 0.16*** | **24.51 ± 0.02*** | **24.31 ± 0.04** | **23.73 ± 0.43** | **24.61 ± 0.14** |
>
> For the LLaMA models, due to resource limitations, we are currently unable to complete 3-seed evaluations before the rebuttal deadline. However, we fully commit to providing the 3-seed results for this model before the camera-ready version if the article is accepted. This will ensure that all claims in the paper are supported by statistically consistent results.
>
> In addition, to enable full reproducibility, we have attached the code with the instructions to run in the submitted supplementary. We will open-source the code if the article is accepted for publication.

---

> > ### Comment · Reviewer_tKGL · 2025-08-04
> >
> > Thank you for addressing most of my concerns, I will maintain the positive score.

---

### Decision · Program_Chairs · 2025-09-17

**Decision:**

Accept (poster)

**Comment:**

This paper proposes a novel methodology for dynamic sparse training in the context of high sparsity regime. The proposed method was studied and demonstrated on various architectures. It is worth highlighting that it obtains impressive results and competitive performance with dense neural networks training for language modelling, a task and architecture (LLaMA family) which is very scarcely studied in the dynamic sparse training literature. The rebuttal addressed most of the reviewers’ concerns. All four reviewers unanimously recommended acceptance. Addressing the concerns raised in the reviews and including the rebuttal discussions in the final version will improve the paper quality even more.